# Multi-view Consistent Latent Action Learning for World Modeling and Control

**Shenghua Wan** [1 2]  **Xiaohai Hu** [3]  **Xunlan Zhou** [1]  **Lei Yuan** [1 2]  **Le Gan** [4]  **De-Chuan Zhan** [1 2]

## Abstract

The scalability of world models is currently bottlenecked by the scarcity of action annotations. While self-supervised latent action learning offers a potential solution, existing single-view paradigms—relying on information bottlenecks or Vector Quantization (VQ)—often conflate superficial 2D pixel displacements with the underlying physical-spatial dynamics of an action. Consequently, these methods remain highly susceptible to view-dependent noise, such as camera shake. We introduce **MuCoLA** (**Mu**lti-view **Co**nsistent **L**atent **A**ction learning), a framework that learns robust, view-invariant action representations by enforcing semantic consistency across synchronized video streams. MuCoLA utilizes a Student-Teacher network with DINO-style self-distillation to align action distributions across viewpoints, effectively filtering high-frequency visual noise while preserving motion semantics. Theoretical analysis reveals that our multi-view objective functions as a spectral filter, isolating agent dynamics from environmental nuisances. Empirically, MuCoLA significantly outperforms baselines in action regression, video reconstruction, and downstream visual control tasks. Furthermore, we demonstrate that MuCoLA exhibits favorable scaling properties with respect to model capacity and data volume, paving the way for large-scale action-free world modeling.

## 1. Introduction

Developing world models to internalize environmental dynamics is a pivotal step toward Artificial General Intelligence (AGI)(Agarwal et al., 2025; Xiang et al., 2025). While autoregressive Transformers (Vaswani et al., 2017) power today's large-scale video generative models(Liu et al., 2024), these approaches often fail to provide the fine-grained action controllability required for effective decision-making (Wu et al., 2024). In response, recent research have turned to learning latent action spaces from video data, a paradigm that shows significant promise for enabling action generalization and cross-embodiment transfer (Ye et al., 2022; Schmidt & Jiang, 2024; Gao et al., 2025).

However, we identify a fundamental limitation in existing single-view paradigms: they primarily observe two-dimensional pixel displacements, often overlooking the underlying spatial-physical semantics of an action (Ye et al., 2022; Gao et al., 2025; Zhang et al., 2025). This makes models prone to conflating an agent's intrinsic motion with transient observation noise, such as camera jitter or fluctuating illumination. Furthermore, current regularization strategies struggle with action complexity: continuous latent spaces constrained by unimodal Gaussian priors $\mathcal{N}(0, I)$ (Gao et al., 2025) are often too simplistic to represent diverse, multi-modal real-world behaviors (e.g., distinct grasping angles), while discrete methods based on vector quantization (VQ) often suffer from low information capacity (Ye et al., 2022; Garrido et al., 2026).

In this work, we propose MuCoLA (Multi-view Consistent Latent Action learning), a framework designed to learn robust, view-invariant action representations. Our core insight is rooted in causality: while visual observations vary dramatically across viewpoints, the underlying physical intervention (the action) driving the state transition remains invariant. Consequently, we propose multi-view consistency as a superior inductive bias. To implement this, we design a Student-Teacher architecture employing cross-view self-distillation inspired by DINO (Caron et al., 2021). Our student network (observing one viewpoint) learns to predict the latent action distribution produced by a momentum teacher (observing a different viewpoint). This cross-view prediction task forces the model to discard view-specific noise and converge on a shared semantic action manifold.

---

[1]School of Artificial Intelligence, Nanjing University, China [2]National Key Laboratory for Novel Software Technology, Nanjing University, China [3]MACS Lab, Department of Mechanical Engineering, University of Washington, The United States [4]School of Computer Science and Technology, Nanjing University of Science and Technology, China. Correspondence to: De-Chuan Zhan <zhandc@nju.edu.cn>.

*Proceedings of the 43rd International Conference on Machine Learning*, Seoul, South Korea. PMLR 306, 2026. Copyright 2026 by the author(s).

Crucially, we theoretically prove that this consistency objective acts as a spectral filter, mathematically isolating agent dynamics from environmental noise.

Our contributions are threefold: (1) We identify the theoretical limitations of Gaussian priors and single-view reconstruction in latent action learning, proposing multi-view consistency as a robust alternative for modeling multi-modal distributions. (2) We introduce MuCoLA, a novel architecture that synergizes continuous latent encoding with discrete prototype clustering via self-distillation. We provide a theoretical analysis showing that this objective effectively filters environmental noise. (3) We demonstrate that world models conditioned on MuCoLA's latent actions achieve state-of-the-art performance in video prediction, visual planning, model-based RL, and robotics control, while exhibiting positive scaling laws with increased data and compute.

## 2. Related Work

**Scalable World Models.** Constructing world models that simulate environmental dynamics is a pivotal step toward general-purpose embodied intelligence. Recent advances have scaled these models by leveraging autoregressive Transformers and massive video datasets. Bruce et al. (2024) introduced a foundation world model trained on Internet gameplay, enabling unsupervised control via spatiotemporal tokens, while Agarwal et al. (2025) and Xiang et al. (2025) developed platforms for physical AI that integrate high-fidelity physics with generative video. To enhance interactivity, Wu et al. (2024) proposed iVideoGPT, which discretizes visual observations into compressed tokens for efficient sequence modeling. However, a critical limitation persists: these models either rely on scarce ground-truth action labels or generate plausible but uncontrollable futures. **MuCoLA** complements this scalable architecture by providing a robust mechanism to learn *latent* control signals from unlabelled multi-view data, bridging the gap between large-scale generative pre-training and precise, actionable decision-making.

**Latent Action Learning.** To circumvent the reliance on explicit action annotations, recent research focuses on learning latent action spaces via information bottlenecks. Early approaches used vector quantization to learn discrete codes from video (Ye et al., 2022; Schmidt & Jiang, 2024; Wang et al., 2024), while subsequent works such as AdaWorld (Gao et al., 2025) advocated continuous latent spaces regularized by Gaussian priors. However, these methods face inherent limitations: continuous spaces under unimodal Gaussian constraints often lack the expressivity to model complex, multi-modal actions, while discrete discretization via VQ tends to provide insufficient information density for precise control (Garrido et al., 2026). Furthermore, Zhang et al. (2025) analyzed linear latent action models and highlighted that single-view reconstruction objectives often conflate high-frequency noise (e.g., camera shake) with agent dynamics. **MuCoLA** addresses these flaws by leveraging a multi-view consistency objective. By enforcing geometric and semantic alignment across views, our approach effectively filters view-dependent noise and captures the intrinsic, multi-modal structure of physical actions.

**Multi-View Learning for Control.** Multi-view learning is an important research field of machine learning (Zhang et al., 2009; Gao et al., 2016; Zhang et al., 2017; Sun et al., 2023), while leveraging multiple viewpoints has proven effective for improving state estimation and robustness in reinforcement learning. Prior work has formalized Multi-View MDPs (Li et al., 2019), leveraging techniques such as masked autoencoding (Seo et al., 2023) and contrastive learning (Kinose et al., 2023) to learn view-invariant state representations. Others have employed teacher-student distillation to transfer privileged multi-view knowledge to single-view policies (Acar et al., 2023). However, these methods predominantly focus on aligning static state representations or requiring multi-view setups during deployment. **MuCoLA** extends these principles to the temporal domain of action learning. Unlike standard MVRL, we distill dynamic action representations, allowing our world model to be trained on rich multi-view data but deployed with a single view, retaining the semantic robustness of the teacher without incurring the inference cost of processing multiple camera streams.

## 3. Method

To capture the complex action distributions inherent in real-world dynamics, we propose **MuCoLA** (Multi-view Consistent Latent Action Learning, as shown in Figure 1), which introduces multi-view geometric constraints as an inductive bias, compelling the model to learn view-invariant representations of physical actions. Our core hypothesis posits that the intrinsic semantics of an agent's physical action must remain consistent regardless of the observational viewpoint. We also provide a theoretical analysis proving that our multi-view consistency objective acts as a noise filter.

### 3.1. Multi-view Consistent Latent Action Modeling

To extract compact and semantically consistent action representations from videos, we design a latent action encoder based on a Spatio-Temporal (ST) Transformer (Peebles & Xie, 2023). Inspired by the architecture of Genie (Bruce et al., 2024) and AdaWorld (Gao et al., 2025), we treat video frames as sequences of spatio-temporal tokens, extending the input formulation to accommodate synchronized multi-view observations.

**Spatio-Temporal Tokenization and Feature Extraction.** Consider a set of multi-view observations at time steps $t$

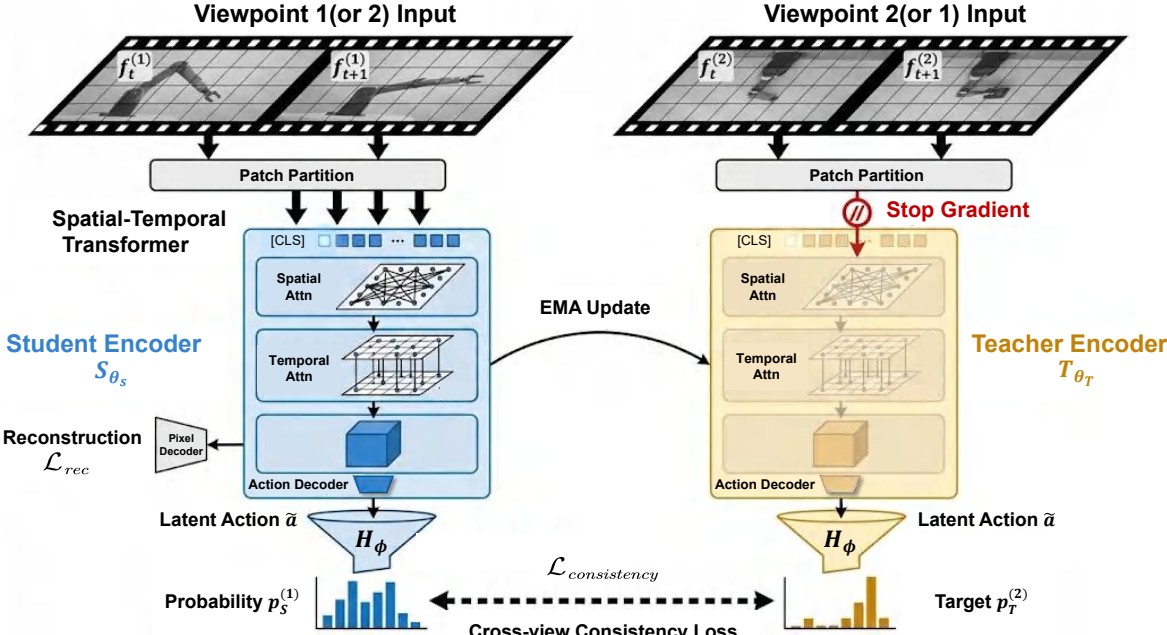

*Figure 1.* **Overview of the MuCoLA Framework.** MuCoLA employs a Student-Teacher architecture to learn view-invariant latent actions. (Left) The Student encoder $S_{\theta_S}$ takes a video sequence from Viewpoint 1, extracts spatio-temporal features, and predicts the latent action distribution. (Right) The momentum Teacher encoder $T_{\theta_T}$ processes the synchronized Viewpoint 2. The model is optimized via a joint objective: a pixel-level reconstruction loss $\mathcal{L}_{rec}$ ensures physical validity, while a cross-view consistency loss $\mathcal{L}_{consistency}$ enforces semantic alignment between the two views via self-distillation.

and $t+1$, denoted as $x = \{(f_t^{(v)}, f_{t+1}^{(v)})\}_{v=1}^V$, where $v$ indicates the view index. For any given view pair, we first partition the frames into non-overlapping patches of size $P \times P$ and project them into $D$-dimensional embeddings via a linear layer. To preserve spatio-temporal structure, we add learnable positional encodings to these tokens. The processed token sequence is then fed into a weight-shared ST-Transformer encoder $E_\theta$. This encoder comprises $L$ stacked layers that interleave Spatial Attention and Temporal Attention mechanisms. Spatial attention captures the visual context within a single frame, while temporal attention models the dynamic transitions across frames. To disentangle action dynamics from visual nuisances (e.g., camera pose, illumination, and occlusion), we prepend a learnable action token, $[CLS]_{act}$, to the input sequence. Through the encoding layers, this token aggregates cross-frame dynamic information and is finally projected to the latent action $\tilde{a}^{(v)} = E_\theta(f_t^{(v)}, f_{t+1}^{(v)})$ specific to that view.

**Cross-View Consistency Constraint.** Our objective is to ensure that $\tilde{a}^{(v)}$ retains only the essential features of the action. Intuitively, for the same physical event, the latent action distribution inferred from view $v_1$ should align with that inferred from view $v_2$. This design overcomes the limitations of single-view priors in modeling multi-modal distributions by implicitly maximizing the mutual information between viewpoints. Consequently, the model is compelled not merely to encode information for pixel recon-

struction but to comprehend the common dynamic features that persist across different perspectives—specifically, the underlying physical actions.

### 3.2. Cross-View Self-Distillation

Directly optimizing the Euclidean distance between the dynamic features of two networks often leads to representation collapse, where all outputs converge to a trivial constant solution. Inspired by DINO (Caron et al., 2021), a self-supervised learning framework, we design a cross-view self-distillation mechanism based on a Teacher-Student architecture to mitigate this issue.

**Model Architecture.** We instantiate a Teacher-Student framework consisting of two parallel networks: a Student network $S_{\theta_S}$ and a Teacher network $T_{\theta_T}$. Both share an identical architecture, comprising the ST-Transformer backbone (from Section 3.1) for extracting the continuous action $\tilde{a}$, and a projection head $H_\phi$. The projection head maps $\tilde{a}$ into a prototype distribution space, outputting a $K$-dimensional probability distribution $p$ that represents the semantic action prototypes.

**Cross-View Consistency Loss.** Instead of a standard contrastive loss, we employ a cross-view cross-entropy loss. Given two views $v_1$ and $v_2$, we train the Student's prediction $p_S^{(1)}$ (conditioned on view $v_1$) to match the Teacher's

target distribution $p_T^{(2)}$ (derived from view $v_2$), and vice versa. This cross-view supervision enforces the learning of view-agnostic semantics. We define the loss function as:

$$\mathcal{L}_{\text{consistency}} = -\frac{1}{2} \sum_k \left[ p_T^{(2)}(k) \log p_S^{(1)}(k) \right. \\ \left. + p_T^{(1)}(k) \log p_S^{(2)}(k) \right] \quad (1)$$

To prevent collapse and enhance training stability, we apply centering and sharpening operations to the Teacher's output. We update the Teacher's parameters $\theta_T$ via an exponential moving average (EMA) of the Student's parameters, i.e., $\theta_T \leftarrow \lambda \theta_T + (1 - \lambda)\theta_S$, with gradients detached (stop-gradient) during backpropagation.

**Total Optimization Objective.** The final loss function of MuCoLA integrates two components: (1) a pixel-level reconstruction loss $\mathcal{L}_{\text{rec}} = \|x - \hat{x}\|_2^2$ (MSE) to ensure the physical validity of latent actions; and (2) the cross-view consistency loss $\mathcal{L}_{\text{consistency}}$ to enforce view invariance.

$$\mathcal{L}_{\text{total}} = \mathcal{L}_{\text{rec}} + \beta \mathcal{L}_{\text{consistency}} \quad (2)$$

Through this joint optimization, the continuous latent action $\tilde{a}$ retains continuous numerical properties for fine-grained control (guaranteed by the reconstruction branch) while acquiring robust semantic consistency.

### 3.3. Action-Aware World Model Training

Upon completing the training of the MuCoLA encoder, we freeze its parameters and utilize it to extract latent action sequences $\tilde{a}_{1:T}$ from the large-scale pretraining dataset.

**Latent-Conditioned Predictive Modeling.** Our world model is trained to maximize the log-likelihood of future observations conditioned on both historical context and extracted latent actions. The optimization objective is formulated as:

$$\mathcal{L}_{wm} = \max_\theta \sum_t \left[ \log p(o_{t+1}|z_{t+1}) + \log p(z_{t+1}|z_{1:t}, \tilde{a}_t) \right] \quad (3)$$

where $z$ denotes the visual state representation and $o$ is the reconstructed observation.

To instantiate this architecture, we adopt iVideoGPT (Wu et al., 2024), a scalable autoregressive Transformer that integrates multimodal signals, including visual observations, rewards, and actions. We first tokenize visual observations $o_t$ using a VQ-GAN and map continuous latent actions $\tilde{a}_t$ to the token embedding space via a linear projection layer $\text{Proj}(\cdot)$. The resulting sequence $\tau$ processed by the Transformer is:

$$\tau = \langle z_1, \text{Proj}(\tilde{a}_1), z_2, \text{Proj}(\tilde{a}_2), \ldots, z_T, \text{Proj}(\tilde{a}_T) \rangle \quad (4)$$

Table 1. Image Reconstruction Metrics on Multi-view Test Sets

| Method | PSNR ↑ | SSIM ↑ | LPIPS% ↓ |
|---|---|---|---|
| AdaWorld (Gao et al., 2025) | | | |
|   View 1 | 34.42 | 96.58 | 13.51 |
|   View 2 | 32.53 | 91.45 | 15.72 |
|   View 1&2, Ave. | 33.53 | 94.01 | 14.63 |
| FICC (Ye et al., 2022) | 30.28 | 90.07 | 18.65 |
| LAPO (Schmidt & Jiang, 2024) | 27.54 | 88.66 | 21.42 |
| **MuCoLA** | **37.97** | **98.28** | **9.42** |
|   w prior | 35.42 | 97.92 | 11.42 |
|   w/o consistency | 32.36 | 93.45 | 14.71 |

This formulation enables the world model to function as a controllable simulator $p(z_{t+1}|z_{1:t}, \tilde{a}_t)$, effectively predicting future visual tokens by attending to past states and current action intentions.

### 3.4. Theoretical Analysis of MuCoLA

To theoretically validate the robustness of our approach, we adopt the *Linear Latent Action Model* (Linear LAM) framework (Zhang et al., 2025). Recent analyses within this framework suggest that single-view reconstruction objectives operate similarly to Principal Component Analysis (PCA) (Abdi & Williams, 2010), often conflating high-variance environmental nuisances (e.g., camera jitter) with agent dynamics. In contrast, we demonstrate that MuCoLA utilizes multi-view consistency as a superior inductive bias, effectively functioning as a spectral filter that disentangles true physical actions from observation noise.

**Proposition 3.1** (Multi-view Consistency as Noise Filtering). *Consider a multi-view setting governed by shared semantics and independent noise. Minimizing the cross-view consistency loss compels the latent action encoders, $D^{(1)}$ and $D^{(2)}$, to project observations onto the subspace of the shared action $\mathbf{a}$, while remaining orthogonal to the subspace of view-specific noise $\boldsymbol{\epsilon}^{(v)}$.*

Intuitively, this proposition asserts that by enforcing alignment across synchronized but distinct viewpoints, the objective maximizes the mutual information related to the agent's intervention. Consequently, the encoder learns to discard view-dependent nuisance factors, isolating the intrinsic geometry of the action manifold. It is important to note that while this linear analysis provides a tractable foundation for understanding the denoising properties of multi-view consistency, our practical implementation utilizes non-linear deep networks and a DINO-style cross-entropy objective. Even so, this theoretical framework offers critical intuitive insight into how the consistency objective serves as a superior inductive bias. We provide the full derivation and proof in Appendix A.

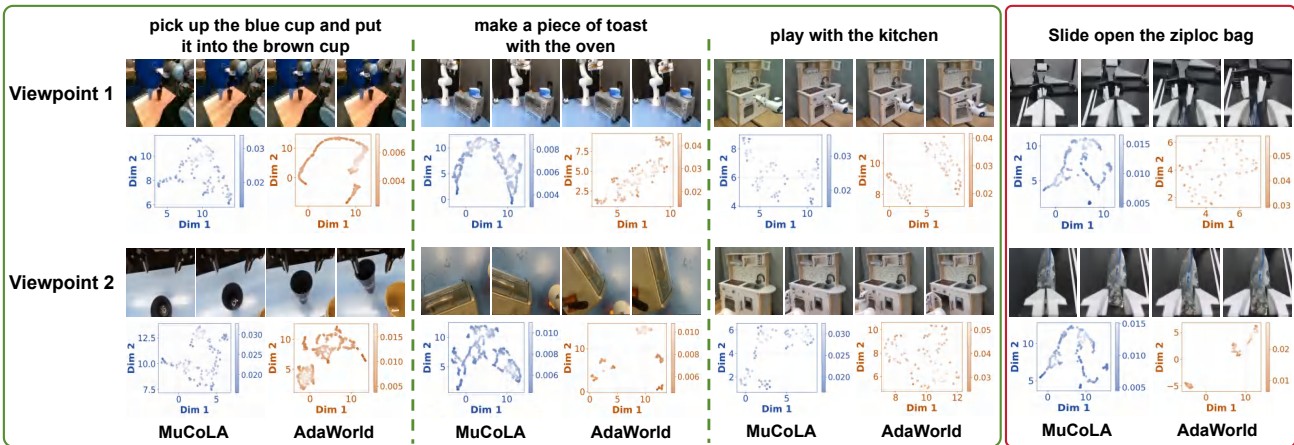

*Figure 2.* **UMAP Visualization of the Learned Latent Action Space.** (Left) Latent actions extracted by MuCoLA and Adaworld from tasks. MuCoLA's Distributions from the training views (Viewpoints 1 and 2) exhibit highly consistent. (Right) These panels demonstrate generalization to a novel perspective. Unlike the training views, Viewpoint 2 here represents an unseen viewpoint (*cam_left_wrist*) used exclusively for evaluation. For brevity, we display only one training view (Viewpoint 1, *cam_high*) as a reference.

*Table 2.* Action Regression Performance Metrics on Datasets.

| Method | RMSE ↓ | | MAE ↓ | |
|---|---|---|---|---|
| | **Train** | **Test** | **Train** | **Test** |
| AdaWorld | $0.229_{\pm0.02}$ | $0.235_{\pm0.07}$ | $0.132_{\pm0.03}$ | $0.135_{\pm0.05}$ |
| FICC | $0.308_{\pm0.03}$ | $0.322_{\pm0.06}$ | $0.203_{\pm0.03}$ | $0.264_{\pm0.09}$ |
| LAPO | $0.264_{\pm0.02}$ | $0.289_{\pm0.04}$ | $0.214_{\pm0.05}$ | $0.231_{\pm0.06}$ |
| **MuCoLA** | $\mathbf{0.202}_{\pm0.02}$ | $\mathbf{0.214}_{\pm0.04}$ | $\mathbf{0.113}_{\pm0.01}$ | $\mathbf{0.120}_{\pm0.05}$ |
| w prior | $0.228_{\pm0.03}$ | $0.233_{\pm0.04}$ | $0.131_{\pm0.02}$ | $0.134_{\pm0.06}$ |
| w/o consistency | $0.220_{\pm0.05}$ | $0.225_{\pm0.10}$ | $0.126_{\pm0.04}$ | $0.129_{\pm0.07}$ |

## 4. Experiments

In this section, we design our experiments to address the following research questions: (**RQ1**) Does MuCoLA capture richer spatial information to enable more precise action characterization (**RQ2**) What characterizes the structure of the latent action space learned by MuCoLA? (**RQ3**) Does MuCoLA demonstrate robust generalization capabilities across different viewpoints? (**RQ4**) Does MuCoLA facilitate downstream tasks, specifically world model learning and control? (**RQ5**) Are the individual designs of MuCoLA effective?

**Datasets.** We train MuCoLA using a diverse collection of multi-source datasets: (1) We utilize 12 datasets from the Open X-Embodiment repository (O'Neill et al., 2024), totaling 9,159 episodes. Each timestep in this collection includes at least two synchronized images captured from significantly different viewpoints. (2) We incorporate the Ego-in-Exo Perception dataset (Reilly et al., 2025) (a subset of Ego-Exo4D (Grauman et al., 2024)), comprising 1,809 videos. This dataset provides authentic paired first-person (egocentric) and third-person (exocentric) viewpoints. (3)

We employ all eight training datasets from the $VP^2$ benchmark (Tian et al., 2023), which contain 22,500 episodes. Since these datasets natively provide only single-viewpoint images, we apply random crop augmentations to the original observations to generate a pseudo-second viewpoint. This ensures the data is compatible with the MuCoLA multi-view training objective. Details are in Appendix D.

**Baselines.** We evaluate our method against three representative baselines with the same spatial-temporal Transformer architecture: FICC (Ye et al., 2022) employs a forward-inverse cycle-consistency objective with vector quantization to implicitly extract discrete action embeddings from action-free videos. LAPO (Schmidt & Jiang, 2024) utilizes a joint inverse-forward dynamics modeling framework to recover latent actions that explain state transitions through an information bottleneck. AdaWorld (Gao et al., 2025) introduces a continuous latent action space optimized via a $\beta$-VAE objective to extract context-invariant representations, facilitating efficient action transfer across diverse environments. Furthermore, we consider two variants of our method: MuCoLA (w/ prior), which imposes an additional Gaussian prior constraint on the generated latent actions, and MuCoLA (w/o consistency), which accepts multi-view observations but omits the consistency loss. We adopt iVideoGPT (Wu et al., 2024) as the backbone world model architecture, which enables a fair comparison of the latent actions generated by all methods across visual reconstruction, future prediction, and model-based RL tasks.

### 4.1. Reconstruction Fidelity and Physical Semantic Representation

We evaluate whether multi-view consistency enhances dynamic representation accuracy. Table 1 demonstrates that

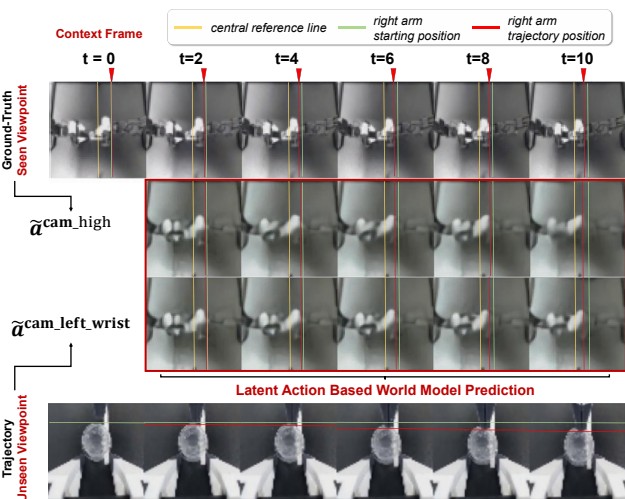

Figure 3. **Cross-View Generalization in World Model Prediction.** We evaluate the world model on the Aloha cup-opening task using latent actions extracted from an unseen left-wrist viewpoint.

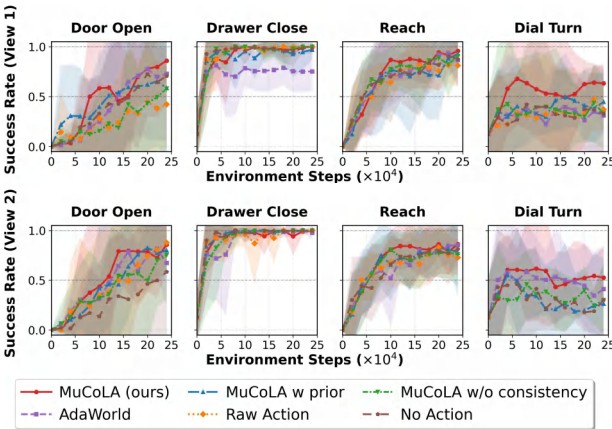

Figure 4. **MBRL Performance on Meta-World.** We evaluate the success rates of agents trained with world model-synthesized data across four manipulation tasks over four seeds. Experiments are conducted on two distinct views: View 1 (rear-side) and View 2 (gripper-centric).

MuCoLA substantially outperforms baseline methods in PSNR (Huynh-Thu & Ghanbari, 2008), SSIM (Wang et al., 2004), and LPIPS (Zhang et al., 2018) metrics for next-frame reconstruction on the datasets, confirming that multi-view consistency effectively filters view-specific noise. Table 2 presents regression results where a two-layer MLP maps latent actions to ground-truth actions on the OpenX dataset (see Appendix B.1). MuCoLA achieves the lowest RMSE and MAE on training and test sets, demonstrating that its latent action space preserves richer physical semantics. In contrast, single-view methods often produce imprecise action representations due to confounding factors such as camera motion.

### 4.2. Latent Action Space Structure

Figure 2 (green region) presents UMAP visualizations of latent actions extracted by MuCoLA across diverse task trajectories. The latent spaces derived from synchronized training viewpoints (Viewpoint 1 and Viewpoint 2) exhibit high structural congruency and distinct semantic clustering. This observation suggests that MuCoLA effectively recovers a shared action manifold where primitive behaviors—such as grasping or pushing—form well-defined, separable regions. In contrast, methods constrained by rigid Gaussian priors (e.g., AdaWorld) often yield diffuse or uninformative distributions that lack explicit semantic partitioning, failing to capture the multi-modal nature of the action space. This result underscores the efficacy of our multi-view consistency objective in inducing a robust, view-invariant, and semantically organized latent representation.

### 4.3. Cross-View Generalization

Figure 2 (red region) demonstrates MuCoLA's generalization to unseen viewpoints: unlike the first three tasks in green region, Viewpoint 2 (*cam_left_wrist*) of the fourth task was never observed during training. Remarkably, latent actions extracted from this novel viewpoint exhibit distribution patterns highly similar to those from training viewpoints after UMAP projection. Figure 3 further validates this capability on the Aloha cup-opening task, where the world model generates consistent future frame predictions using latent actions from an unseen left-wrist viewpoint. The model correctly anticipates the leftward motion of the robotic arm, matching both ground-truth trajectories and predictions from seen viewpoints. Crucially, MuCoLA captures the essential physical semantics of actions rather than superficial visual changes, as the same action appears with different visual flows across viewpoints. This confirms the superiority of our multi-view alignment mechanism for learning viewpoint-invariant action representations.

### 4.4. Downstream Planning and Control

We evaluate the utility of the pre-trained world model in diverse downstream tasks.

**Visual Planning.** We evaluate Model Predictive Control (MPC) performance on the VP$^2$ benchmark (Tian et al., 2023). To enable physical execution, we ground the latent actions $\tilde{a}$ to robot actuators $a_{real}$ via a lightweight two-layer MLP (see Appendix B.4). Using this mapping, we train the iVideoGPT backbone on trajectories from Robosuite and RoboDesk. As shown in Figure 5, MuCoLA consistently outperforms AdaWorld and attains parity with the oracle *Raw Action* model. Notably, MuCoLA achieves state-of-the-

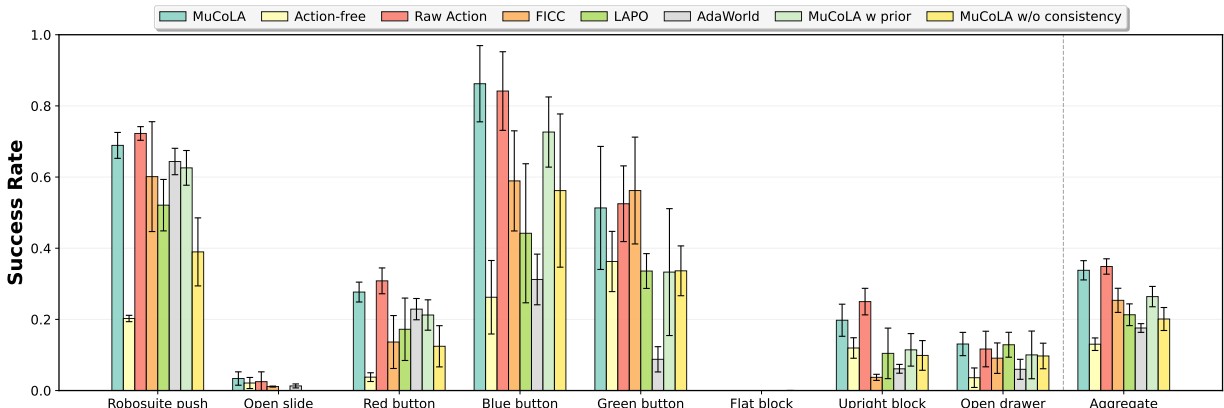

*Figure 5.* **Visual Planning Success Rates on VP²Benchmark.** We compare the Model Predictive Control (MPC) performance of world models conditioned on latent actions from MuCoLA and baselines across 8 tasks, each over 100 runs. MuCoLA (green) consistently outperforms baseline methods.

*Table 3.* **Behavior Cloning Success Rates on LIBERO Benchmark.** Comparison of average success rates ($\pm$ standard deviation) across four task suites over 50 runs.

| Method | Long | Goal | Object | Spatial |
|---|---|---|---|---|
| MuCoLA | **65.4±10.1%** | 73.8±3.1% | **45.3±13.1%** | **59.2±8.3%** |
| AdaWorld | 55.2±4.9% | **76.1±8.3%** | 40.2±4.9% | 52.9±4.5% |
| LAPO | 58.0±8.3% | 62.1±19.0% | 33.9±2.1% | 42.7±6.8% |

art success rates on the "Blue Button" task, underscoring the robustness of our latent control signals. Qualitatively, Figure 6 reveals that while baselines suffer from visual artifacts or task failure, MuCoLA generates high-fidelity futures that closely track the ground-truth trajectory.

**Model-Based RL.** Following the iVideoGPT framework (Wu et al., 2024), we adopt a model-based reinforcement learning approach that augments the experience replay buffer with synthetic trajectories generated by the world model to train a standard Actor-Critic agent (see Appendix B.5). We evaluate methods on four robotic manipulation tasks from the Meta-World benchmark (Yu et al., 2020), using data from two distinct perspectives (View 1: rear-side; View 2: gripper-centric). In Figure 4, MuCoLA consistently outperforms baseline methods across both View 1 and View 2. These results indicate that world models conditioned on multi-view consistent latent actions significantly enhance downstream RL performance across diverse visual settings.

**Behavior Cloning.** We evaluate our method on the LIBERO benchmark (Liu et al., 2023), comprising 40 tasks across four suites (Long, Goal, Object, and Spatial). Leveraging the provided expert demonstrations, we infer latent actions $\tilde{a}$ from image observations using MuCoLA and baseline methods. We adopt a ViT-T backbone from LIBERO benchmark (Liu et al., 2023) as the policy network, training a behavior cloning agent $\pi$ that maps observations and task

instructions to these inferred latent actions. To ground these latents in physical control, we train a lightweight action decoder using a small set of ground-truth action labels (more details are in Appendix B.6). As summarized in Table 3, MuCoLA outperforms the AdaWorld and LAPO baselines in average success rates across the Long, Object, and Spatial suites. These findings, corroborating the reconstruction analysis in Section 4.1, demonstrate that MuCoLA captures superior physical action semantics, thereby enabling more robust performance in downstream manipulation tasks.

### 4.5. Ablation Study and Scalability Analysis

**Impact of Gaussian Priors.** To evaluate the necessity of our learned distribution, we introduce an explicit KL divergence constraint to enforce a Gaussian prior on the latent action space (with a loss weight of 0.5). As shown in Table 1, this constraint degrades image reconstruction quality across all metrics. Similarly, regression performance against ground-truth actions deteriorates compared to the original MuCoLA (Table 2). Downstream evaluations on visual planning (Figure 5) and MBRL tasks (Figure 4) reveal a corresponding performance drop. We attribute this to the limited expressivity of unimodal Gaussian priors, which fail to adequately model the multi-modal nature of complex action trajectories inherent in physical dynamics.

**Efficacy of Cross-View Consistency.** To demonstrate that our performance gains stem from the proposed consistency mechanism rather than merely the ingestion of multi-view data, we ablate the cross-view consistency loss. This modification leads to a sharp decline in reconstruction fidelity (Table 1) and action regression accuracy (Table 2). Furthermore, performance on downstream visual planning (Figure 5) and MBRL tasks (Figure 4) falls significantly below that of the full MuCoLA model. These results underscore the critical role of the consistency objective in filtering view-dependent

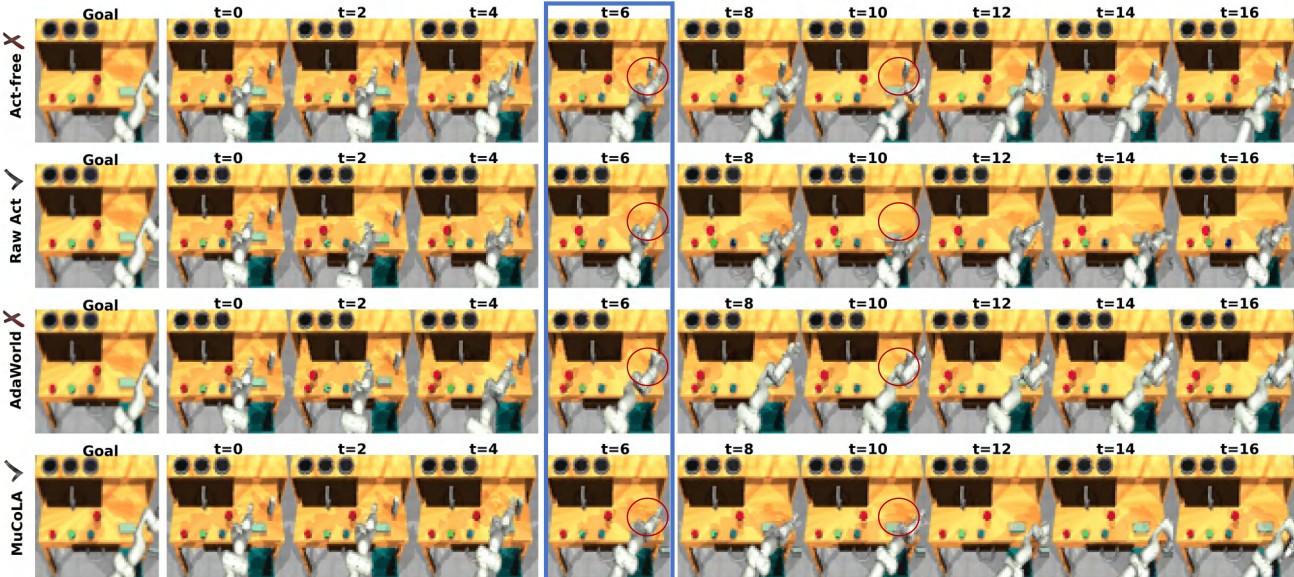

*Figure 6.* **Qualitative Comparison of Visual Planning Trajectories.** Visualizations of open-loop plans in the *upright_block_off_table* task generated by world models conditioned on different action representations. Blue boxes mark the key frames of robotic arm–object contact, while red circles highlight the contact locations across different methods.

noise and extracting robust, semantically meaningful latent actions, which aligns with the theoretical analysis.

**Hyper-parameters Sensitivity.** To address concerns regarding the cost of grounding latent actions, Figure 7(a) evaluates the success rate of the downstream policy on LIBERO as a function of available ground-truth action labels. While performance naturally correlates with label quantity, MuCoLA achieves robust success rates with only 5K labeled samples. Even in the low-data regime (1K samples), the model retains non-trivial control capabilities, underscoring that our semantically structured latent space significantly lowers the sample complexity for downstream physical grounding. We further investigate the robustness of MuCoLA with respect to the discretization granularity $K$. In Figure 7(b), We observe that smaller vocabularies ($K < 2048$) yield higher reconstruction errors, likely due to an inability to cover the highly multi-modal action distribution inherent in manipulation tasks. Conversely, an excessively large vocabulary ($K = 8192$) leads to a slight performance degradation, potentially due to cluster fragmentation or sparse gradient signals during training.

**Scalability Analysis.** We investigate the scaling properties of MuCoLA by varying model capacity and dataset size. Increasing the parameter count of the Spatial-Temporal Transformer results in a monotonic reduction in LPIPS metrics for next-frame reconstruction (Figure 7(c)). A similar trend is observed when scaling the training data from 1% to 100% of the original dataset size (Figure 7(d)). These findings confirm that MuCoLA benefits effectively from scaling laws, exhibiting continuous performance improvements with in-

creased computational resources and data volume.

## 5. Conclusion

We presented MuCoLA, a framework that rethinks latent action learning through the lens of multi-view consistency. By learning view-invariant distribution via self-distillation, MuCoLA successfully captures the underlying physical interventions driving the state transition. Our theoretical analysis and extensive empirical evaluation confirm that this approach yields latent actions that are both geometrically precise and semantically robust. Notably, MuCoLA exhibits favorable scaling properties, showing continuous improvement with larger model capacities and dataset sizes.

Despite these advances, the broad applicability of MuCoLA remains currently constrained by its reliance on synchronized multi-view video datasets. While recent efforts such as Ego-Exo4D (Grauman et al., 2024) provide high-quality multi-view perspectives of human activities, these collections remain significantly smaller in scale than the vast, uncurated video repositories available on the internet. Future work will focus on the continued curation of large-scale, multi-view natural datasets and, more importantly, on scaling MuCoLA to massive internet-scale video via advanced view-augmentation and pseudo-view synthesis techniques. We believe that multi-view consistency represents a scalable and essential inductive bias for the next generation of general-purpose world models, providing the grounding necessary for truly robust embodied intelligence.

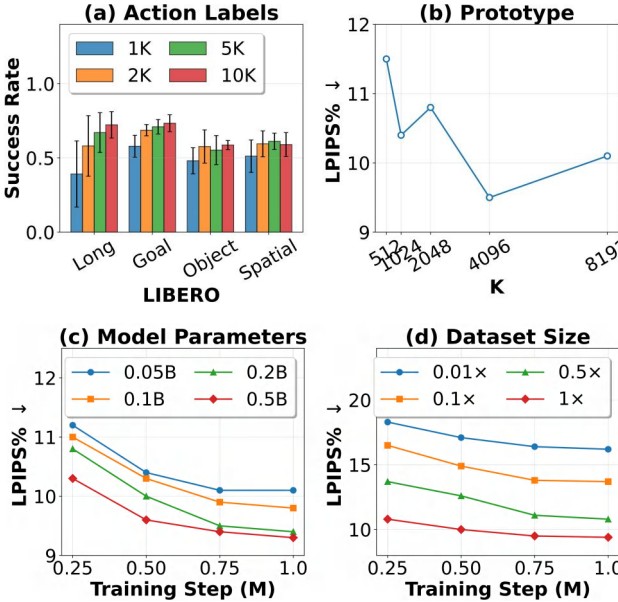

*Figure 7.* **Ablation and Scalability Analysis.** We systematically investigate the impact of four key factors on MuCoLA's performance: downstream label efficiency (a), prototype granularity $K$ (b), and scalability with respect to model parameters (c) and training data volume (d).

## Impact Statement

This paper presents work aimed at advancing the field of Machine Learning. There are many potential societal consequences of our work, none of which we feel must be specifically highlighted here.

## Acknowledgements

This work was partially supported by the Young Scientists Fund of the National Natural Science Foundation of China (Ph.D Candidate) under Grant No. 624B200197, the National Science and Technology Major Project under Grant No. 2022ZD0114805, Collaborative Innovation Center of Novel Software Technology and Industrialization, NSFC (62476123, 62376118, 62006112, 62250069, 61921006, 62495090, 62495094), the Fundamental and Interdisciplinary Disciplines Breakthrough Plan of the Ministry of Education of China (No. JYB2025XDXM118), the "111 Center" (No. B26023), and the Collaborative Innovation Center of Novel Software Technology and Industrialization.

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

# A. Theoretical Analysis: Multi-view Consistency as Spectral Filtering

In this section, we provide a theoretical justification for MuCoLA using the *Linear Latent Action Model* (Linear LAM) framework recently proposed by Zhang et al.. While practical world models employ deep neural networks, the linear analysis captures the essential dynamics of learning and offers tractable insights into why multi-view consistency serves as a superior inductive bias compared to single-view reconstruction, particularly in the presence of environmental noise.

## A.1. Setup and Definitions

Following the formulation in Linear LAM (Zhang et al., 2025), we model the transition between observations as a linear combination of controllable actions and exogenous noise. We extend this to a multi-view setting.

**Data Generation Process.** Consider a set of synchronized multi-view observations. For a view $v \in \{1, 2\}$, the observation change $\Delta\mathbf{o}^{(v)} = \mathbf{o}'^{(v)} - \mathbf{o}^{(v)}$ is defined as:

$$\Delta\mathbf{o}^{(v)} = X^{(v)}\mathbf{a} + \boldsymbol{\epsilon}^{(v)} \tag{5}$$

where:

- $\mathbf{a} \sim \mathcal{N}(\mathbf{0}, I_{d_a})$ is the ground-truth *latent action* (e.g., robot control), which is the shared causal factor across views.

- $X^{(v)}$ is the view-specific *action effect matrix* mapping the action to visual changes in view $v$.

- $\boldsymbol{\epsilon}^{(v)} \sim \mathcal{N}(\mathbf{0}, \Sigma_\epsilon^{(v)})$ represents view-specific *exogenous noise* (e.g., camera jitter, lighting flicker).

**Key Assumptions.** To analyze the effectiveness of consistency, we introduce two standard assumptions regarding the multi-view structure:

1. **Shared Semantics:** The action $\mathbf{a}$ is the only common cause of dynamics across views.

2. **Independent Noise:** The noise is independent across views and independent of the action. That is, $\mathbb{E}[\boldsymbol{\epsilon}^{(1)}(\boldsymbol{\epsilon}^{(2)})^T] = \mathbf{0}$ and $\mathbb{E}[\mathbf{a}(\boldsymbol{\epsilon}^{(v)})^T] = \mathbf{0}$.

**Linear Encoder.** We define the linear latent action encoder for view $v$ as a matrix $D^{(v)}$ that maps the observation difference to the latent representation $\mathbf{z}^{(v)}$:

$$\mathbf{z}^{(v)} = D^{(v)}\Delta\mathbf{o}^{(v)} = D^{(v)}(X^{(v)}\mathbf{a} + \boldsymbol{\epsilon}^{(v)}) \tag{6}$$

## A.2. Limitation of Single-View Learning (PCA)

As derived in Proposition 4.1 of (Zhang et al., 2025), optimizing a single-view reconstruction objective is equivalent to performing Principal Component Analysis (PCA) on the transition distribution. The objective effectively maximizes the variance of the learned latent $\mathbf{z}$:

$$\max_D \mathrm{Var}(\mathbf{z}) = \max_D \mathbb{E}[\|D(X\mathbf{a} + \boldsymbol{\epsilon})\|^2] \tag{7}$$

**Failure Mode:** If the variance of the noise dominates the signal (i.e., $\lambda_{\max}(\Sigma_\epsilon) \gg \lambda_{\max}(XX^T)$), the encoder $D$ will align with the principal components of the noise $\boldsymbol{\epsilon}$ rather than the action $\mathbf{a}$. Consequently, the learned latent space captures nuisance factors (e.g., camera shake) instead of physical actions.

## A.3. Derivation of MuCoLA's Denoising Property

MuCoLA introduces a cross-view consistency constraint. In the linear setting, this is analogous to minimizing the Euclidean distance between latent representations from different views.

**Proposition A.1** (Multi-view Consistency as Noise Filtering). *Under the assumptions of shared semantics and independent noise, minimizing the cross-view consistency loss forces the latent action encoders $D^{(1)}, D^{(2)}$ to project observations onto the subspace of the shared action $\mathbf{a}$ and orthogonal to the subspace of view-specific noise $\boldsymbol{\epsilon}^{(v)}$.*

*Proof.* The consistency loss function $\mathcal{L}_{\text{consist}}$ is defined as the expected squared difference between view embeddings:

$$\mathcal{L}_{\text{consist}} = \mathbb{E}_{\mathbf{a}, \boldsymbol{\epsilon}} \left[ \|\mathbf{z}^{(1)} - \mathbf{z}^{(2)}\|^2 \right] \tag{8}$$

Substituting the definition of $\mathbf{z}^{(v)}$:

$$\mathcal{L}_{\text{consist}} = \mathbb{E} \left[ \|(D^{(1)}X^{(1)}\mathbf{a} + D^{(1)}\boldsymbol{\epsilon}^{(1)}) - (D^{(2)}X^{(2)}\mathbf{a} + D^{(2)}\boldsymbol{\epsilon}^{(2)})\|^2 \right] \tag{9}$$

We group the terms by sources (action $\mathbf{a}$ and noise $\boldsymbol{\epsilon}$):

$$\mathcal{L}_{\text{consist}} = \mathbb{E} \left[ \| \underbrace{(D^{(1)}X^{(1)} - D^{(2)}X^{(2)})\mathbf{a}}_{\text{Signal Mismatch}} + \underbrace{D^{(1)}\boldsymbol{\epsilon}^{(1)}}_{\text{Noise View 1}} - \underbrace{D^{(2)}\boldsymbol{\epsilon}^{(2)}}_{\text{Noise View 2}} \|^2 \right] \tag{10}$$

Expanding the squared norm $\|\mathbf{u} + \mathbf{v} + \mathbf{w}\|^2 = \|\mathbf{u}\|^2 + \|\mathbf{v}\|^2 + \|\mathbf{w}\|^2 + $ cross terms. Due to the independence assumptions ($\mathbb{E}[\mathbf{a}\boldsymbol{\epsilon}^T] = 0$ and $\mathbb{E}[\boldsymbol{\epsilon}^{(1)}\boldsymbol{\epsilon}^{(2)T}] = 0$), all cross-terms vanish in expectation. The loss simplifies to:

$$\mathcal{L}_{\text{consist}} = \underbrace{\mathbb{E}[\|(D^{(1)}X^{(1)} - D^{(2)}X^{(2)})\mathbf{a}\|^2]}_{\text{Term A: Semantic Alignment}} + \underbrace{\mathbb{E}[\|D^{(1)}\boldsymbol{\epsilon}^{(1)}\|^2]}_{\text{Term B: Noise Suppression 1}} + \underbrace{\mathbb{E}[\|D^{(2)}\boldsymbol{\epsilon}^{(2)}\|^2]}_{\text{Term C: Noise Suppression 2}} \tag{11}$$

To minimize $\mathcal{L}_{\text{consist}}$, the model must simultaneously minimize all three positive terms:

1. Minimizing **Term B** and **Term C** requires $D^{(v)}\boldsymbol{\epsilon}^{(v)} \to \mathbf{0}$. This implies that the encoder $D^{(v)}$ must be effectively **orthogonal** to the noise subspace (i.e., $D^{(v)}$ acts as a spectral filter rejecting $\boldsymbol{\epsilon}^{(v)}$).

2. Minimizing **Term A** requires $D^{(1)}X^{(1)} \approx D^{(2)}X^{(2)}$, enforcing that the extracted latents represent the shared geometry of the action $\mathbf{a}$.

$\square$

**Remark.**  Equation 11 theoretically validates why MuCoLA outperforms baselines in noisy environments. While single-view methods (PCA-like) are biased towards high-variance noise, MuCoLA (resembling Canonical Correlation Analysis, CCA (Thompson, 1984)) exploits the independence of noise across views to disentangle and discard it. Note that while our theoretical analysis assumes a linear model with Euclidean distance for tractability, our practical implementation employs a DINO-style cross-entropy objective with centering and sharpening. This modification prevents representation collapse in high-dimensional deep feature spaces while preserving the geometric alignment properties derived above.

## B. Implementation Details and Hyperparameters

### B.1. Training Details of Latent Action Model

The MuCoLA architecture is instantiated as a Spatio-Temporal Transformer (Peebles & Xie, 2023). The encoder processes video clips consisting of $T = 2$ frames with a resolution of $128 \times 128$. We utilize a patch size of $16 \times 16$, resulting in a sequence of patch embeddings projected to a model dimension of 1024. The architecture comprises 8 stacked encoder blocks and 8 decoder blocks, each utilizing 16 attention heads. The continuous latent action dimension is set to $d = 32$.

For the discretization bottleneck involved in the consistency objective, we set the prototype dimension to $K = 4096$. We employ a DINO-style self-distillation loss (Caron et al., 2021) with a weight of $\beta = 1.0$. The student and teacher temperatures are set to $\tau_s = 0.1$ and $\tau_t = 0.07$, respectively, with a center momentum of 0.9 and a teacher momentum schedule of $\lambda = 0.996$.

The model is trained using the AdamW optimizer with a learning rate of $2.5 \times 10^{-5}$ and a weight decay of $1 \times 10^{-2}$. We use a batch size of 60 per GPU. The entire training process is conducted on a compute node equipped with 8 NVIDIA RTX 4090 GPUs and requires approximately 48 hours to converge.

To enable downstream physical execution, we train a lightweight projection head (two-layer MLPs) to map the learned latent actions $\tilde{a}$ to ground-truth robot actions $a_{gt}$ using a small subset of labeled data.

### B.2. Training Details of World Model

We adopt the iVideoGPT architecture (Wu et al., 2024) as the backbone for our world model. The visual tokenizer is a VQ-GAN (Esser et al., 2021) with a codebook size of $8192$ and a latent channel dimension of $64$, compressing $128 \times 128$ images into $32 \times 32$ discrete tokens.

The autoregressive dynamics model is based on the LLaMA architecture (Touvron et al., 2023), configured with 12 hidden layers, 12 attention heads, and a hidden size of 768. We utilize the SiLU activation function and Rotary Positional Embeddings (RoPE) (Su et al., 2024). The model is trained to predict the next token sequence conditioned on historical visual tokens and action tokens. We compare three action conditioning variants: (1) **Raw Action**, using the original actuator dimensions; (2) **Action-free**, utilizing no action conditioning; and (3) **Latent Action**, utilizing MuCoLA's 32-dimensional continuous latent codes projected into the transformer's embedding space.

### B.3. Inference and Deployment from Unseen Viewpoints

A key capability of MuCoLA is cross-view generalization. During the training of the world model, we utilize synchronized multi-view data (e.g., 'cam_high' and 'cam_right_wrist' for the *Aloha* tasks). During inference, we evaluate the system based on latent actions from a strictly unseen viewpoint (e.g., 'cam_left_wrist').

We first employ the frozen MuCoLA encoder to infer the latent action $\tilde{a}$ from the unseen viewpoint observations. This inferred latent action is then paired with the visual observation from a seen viewpoint (e.g., 'cam_high') and fed into the world model. This setup tests whether the semantic control signal extracted from a novel perspective is compatible with the dynamics learned from familiar perspectives.

### B.4. Training and Evaluation on VP$^2$

The VP$^2$ benchmark (Tian et al., 2023) consists primarily of single-view datasets, which presents a challenge for MuCoLA's multi-view consistency objective. To bridge this gap, we employ a pseudo-multiview generation strategy. For every pair of adjacent frames $f_t^{(1)}, f_{t+1}^{(1)}$, we generate a synchronized pseudo-view $f_t^{(2)}, f_{t+1}^{(2)}$ via spatial augmentation, specifically applying a random crop with an 8-pixel offset. This forces the model to learn invariance to local spatial shifts.

Simultaneously, we train a forward mapper $D^\star : a \to \tilde{a}$ that predicts latent actions from ground-truth actions, which implemented as two-layer MLP networks. During the world model training phase, we condition predictions on the original observations and the inferred latent actions $\tilde{a}$. For evaluation, given an initial frame $f_0$ and a sequence of ground-truth actions $a_{1:T}$, we first map the actions to the latent space using $D^\star$ to obtain $\tilde{a}_{1:T}$. The world model then predicts the future trajectory based on these latent codes, and the synthesized frames are passed to a pre-trained discriminator to estimate the success rate.

### B.5. Training details of MBRL tasks

We adopt the Model-Based Policy Optimization (MBPO) framework (Janner et al., 2019) to train our agent following the paradigm in iVideoGPT (Wu et al., 2024). The training process involves three primary components: an actor-critic policy $(\pi_\phi, v_\psi)$, a learned world model $p_\theta$, and two experience replay buffers—a real buffer $\mathcal{D}_{\text{real}}$ and an imagined buffer $\mathcal{D}_{\text{imag}}$.

First, we initialize $\mathcal{D}_{\text{real}}$ by collecting trajectories using a random policy and pre-train the world model $p_\theta$ on this dataset to establish an initial understanding of the environment dynamics. $\mathcal{D}_{\text{imag}}$ is initialized with 10 random rollouts generated by the world model. The main training loop proceeds for $N = 250,000$ environment steps. In each iteration, we perform model learning by updating $p_\theta$ to maximize the log-likelihood of transitions sampled from a mini-batch of 256 transitions from $\mathcal{D}_{\text{real}}$.

Simultaneously, we update the actor-critic networks using model-free objectives computed on a mixture of data from both $\mathcal{D}_{\text{real}}$ and $\mathcal{D}_{\text{imag}}$. To populate the imagined buffer, we periodically execute model rollouts: we sample initial states $o_t$ uniformly from $\mathcal{D}_{\text{real}}$ and perform $k$-step latent rollouts using the current policy $\pi_\phi$, adding the resulting synthetic trajectories to $\mathcal{D}_{\text{imag}}$. For data collection in the real environment, the agent executes actions selected by $\pi_\phi$ with an action repeat of 2, and the resulting transitions are stored in $\mathcal{D}_{\text{real}}$.

*Table 4.* **MuCoLA Training Hyperparameters (Stage 1)**

| Hyperparameter | Value |
| --- | --- |
| *Model Architecture* | |
| Input Resolution | $128 \times 128$ |
| Patch Size | $16 \times 16$ |
| Encoder/Decoder Blocks | 8 |
| Model Dimension | 1024 |
| Num Heads | 16 |
| Latent Action Dim | 32 |
| Prototype Dim ($K$) | 4096 |
| *Optimization & DINO* | |
| Optimizer | AdamW |
| Learning Rate | 2.5e-5 |
| Weight Decay | 1e-2 |
| Batch Size | 60 |
| Frames per Clip | 2 |
| DINO Loss Weight ($\beta$) | 1.0 |
| Student Temperature | 0.1 |
| Teacher Temperature | 0.07 |
| Center Momentum | 0.9 |
| Teacher Momentum ($\lambda$) | 0.996 |

### B.6. Training and Evaluation on LIBERO

For the LIBERO benchmark (Liu et al., 2023), we evaluate the utility of latent actions for Behavior Cloning (BC). We adopt a two-stage training process to ensure fair comparison. First, we use the pre-trained MuCoLA (or baseline) encoder to infer latent actions $\tilde{a}$ from the expert demonstrations. We then train a policy network $\pi$ (based on a ViT-T backbone) to predict these latent actions from visual observations, i.e., $\tilde{a}_{\text{pred}} = \pi(o, \text{instruction})$.

To enable execution in the simulation, we concurrently train and freeze a separate decoder module that maps the latent action $\tilde{a}$ back to the physical robot action space $a_{\text{real}}$. During evaluation, the policy predicts a latent action, which is then decoded into a physical control signal to interact with the environment. Success is measured by the average task completion rate over the evaluation episodes.

### B.7. Hyperparameters

We provide the detailed hyperparameter settings for reproducing our results. Table 4 lists the parameters for the MuCoLA latent action learning stage (Stage 1), and Table 5 details the configuration for the World Model training (Stage 2).

*Table 5.* **World Model Hyperparameters (Stage 2)**

| Hyperparameter | Value |
| --- | --- |
| *VQ-GAN / Visual Tokenizer* | |
| Codebook Size | 8192 |
| Latent Channels | 64 |
| Downsample Factor | 4 (Resolution 64) |
| *Transformer Backbone* | |
| Block Out Channels | [128, 256, 512] |
| Layers Per Block | 2 |
| Norm Type | Group Norm (32 groups) |
| Activation | SiLU |
| Context Length | 2 |

---

**Algorithm 1** MuCoLA Training Pipeline

---

**Require:** Multi-view video dataset $\mathcal{D}$, Batch size $B$, EMA decay $\lambda$, Loss weight $\beta$
**Require:** Student encoder $S_{\theta_S}$, Teacher encoder $T_{\theta_T}$, Projection heads $H_{\phi_S}, H_{\phi_T}$
 1: **Initialize:** $\theta_S, \phi_S$ randomly; $\theta_T \leftarrow \theta_S, \phi_T \leftarrow \phi_S$
 2: *// Stage 1: Multi-view Consistent Latent Action Learning*
 3: **while** not converged **do**
 4:    Sample batch of synchronized views $\{(x^{(1)}, x^{(2)})_i\}_{i=1}^B$ from $\mathcal{D}$
 5:    *// Student Forward (Gradients enabled)*
 6:    $\tilde{a}_S^{(1)}, \tilde{a}_S^{(2)} \leftarrow S_{\theta_S}(x^{(1)}), S_{\theta_S}(x^{(2)})$ *// Extract continuous latent actions*
 7:    $p_S^{(1)}, p_S^{(2)} \leftarrow H_{\phi_S}(\tilde{a}_S^{(1)}), H_{\phi_S}(\tilde{a}_S^{(2)})$ *// Predict semantic prototypes*
 8:    *// Teacher Forward (Stop Gradient)*
 9:    **with** torch.no_grad():
10:       $\tilde{a}_T^{(1)}, \tilde{a}_T^{(2)} \leftarrow T_{\theta_T}(x^{(1)}), T_{\theta_T}(x^{(2)})$
11:       $p_T^{(1)}, p_T^{(2)} \leftarrow \text{CenterSharp}(H_{\phi_T}(\tilde{a}_T^{(1)})), \text{CenterSharp}(H_{\phi_T}(\tilde{a}_T^{(2)}))$
12:    *// Compute Objectives*
13:    Calculate reconstruction loss (using pixel decoder):
14:    $\mathcal{L}_{\text{rec}} = \mathcal{L}_{\text{pixel}}(x^{(1)}, \tilde{a}_S^{(1)}) + \mathcal{L}_{\text{pixel}}(x^{(2)}, \tilde{a}_S^{(2)})$
15:    Calculate cross-view consistency loss (Eq. 1):
16:    $\mathcal{L}_{\text{consistency}} = -\frac{1}{2} \sum_k \left[ p_T^{(2)}(k) \log p_S^{(1)}(k) + p_T^{(1)}(k) \log p_S^{(2)}(k) \right]$
17:    $\mathcal{L}_{\text{total}} = \mathcal{L}_{\text{rec}} + \beta \mathcal{L}_{\text{consistency}}$
18:    *// Optimization*
19:    Update Student parameters $\theta_S, \phi_S$ via Adam on $\nabla \mathcal{L}_{\text{total}}$
20:    Update Teacher via EMA: $\theta_T \leftarrow \lambda\theta_T + (1-\lambda)\theta_S, \phi_T \leftarrow \lambda\phi_T + (1-\lambda)\phi_S$
21: **end while**
22:
23: *// Stage 2: Latent-Action-Aware World Model Training*
24: Freeze Latent Action Encoder $S_{\theta_S}$
25: **while** not converged **do**
26:    Sample trajectory $\tau$ from $\mathcal{D}$
27:    Extract latent actions $\tilde{a}_t = S_{\theta_S}(x_t)$ for all steps
28:    Tokenize visual observations $o_t \rightarrow z_t$
29:    Maximize log-likelihood $\mathcal{L}_{wm}$ (Eq. 3) for next-token prediction
30: **end while**

---

## C. Pseudo Codes

To provide a clear and comprehensive understanding of the MuCoLA algorithm, we present the pseudocode in Algorithm 1. We have uploaded the source code for MuCoLA in the supplementary material.

## D. Datasets Details

Table 6 summarizes the datasets employed in our experiments. The "Training Viewpoints" column identifies the synchronized dual-view pairs utilized during the training phase, whereas "Additional Viewpoints" lists the unseen perspectives reserved for assessing the cross-view generalization capabilities of our latent action model (Section 4.3). We reweight the sampling rates for different datasets according to the number of frames in each dataset. We provide qualitative visualizations for each dataset in Figures 8, 9, and 10.

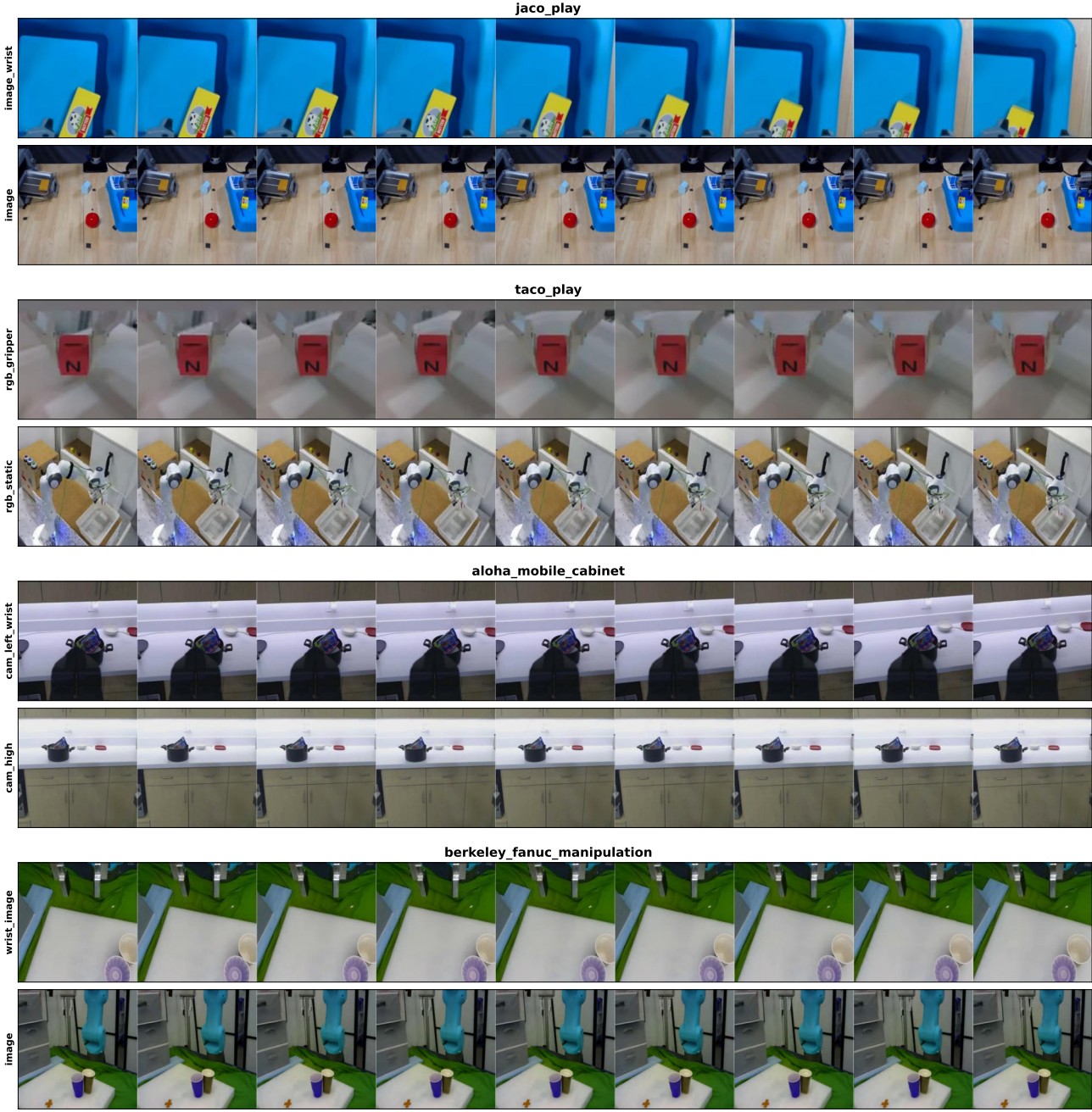

*Figure 8.* Sampled trajectories of first-person (bottom) and third-person (top) views across robotics datasets (aloha_mobile_cabinet, berkeley_fanuc_manipulation, jaco_play, taco_play) from OpenX-Embodiment.

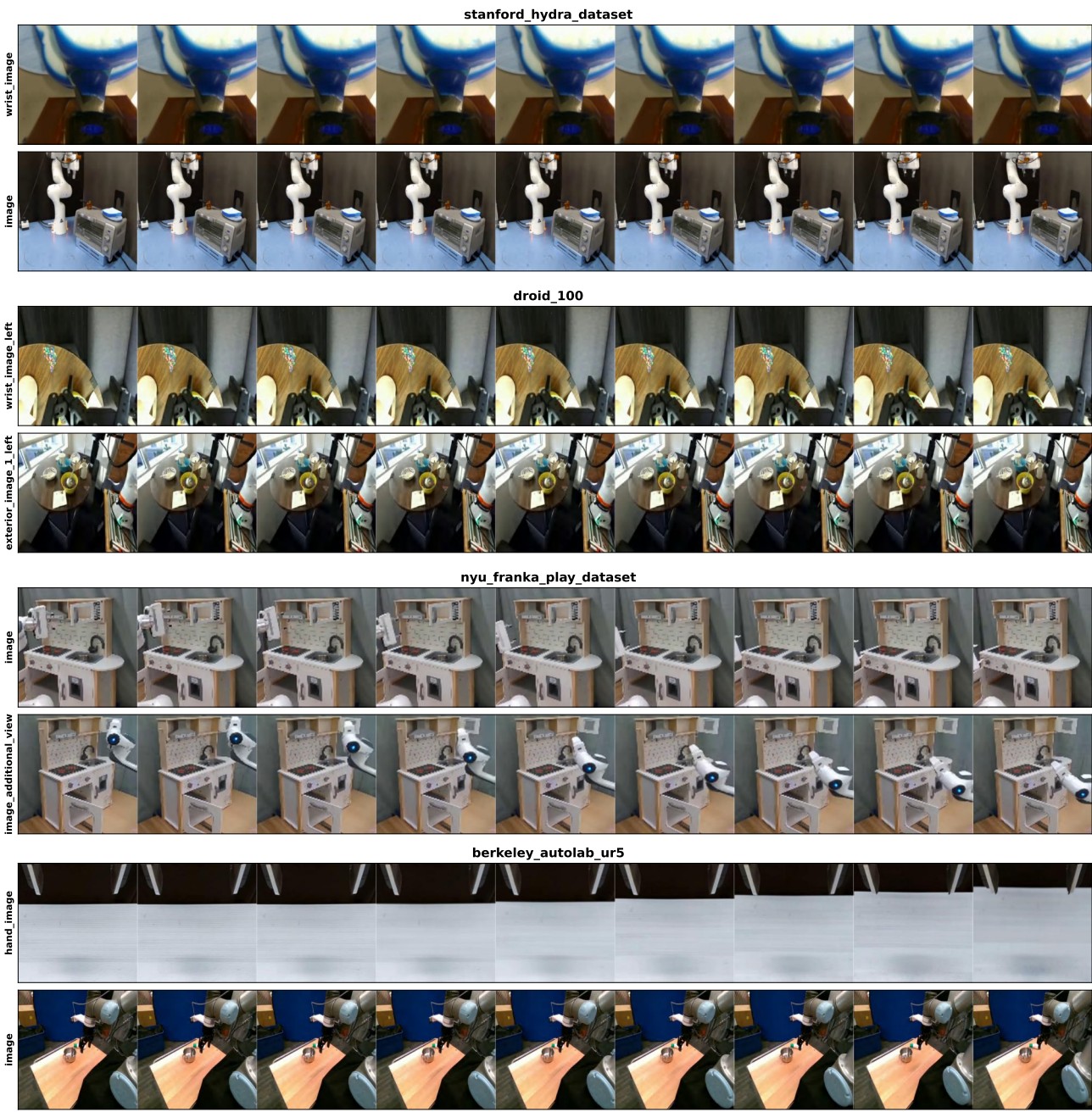

*Figure 9.* Sampled trajectories of first-person (bottom) and third-person (top) views across robotics datasets (stanford_hydra_dataset, droid_100, nyu_franka_play_dataset, berkeley_autolab_ur5) from OpenX-Embodiment.

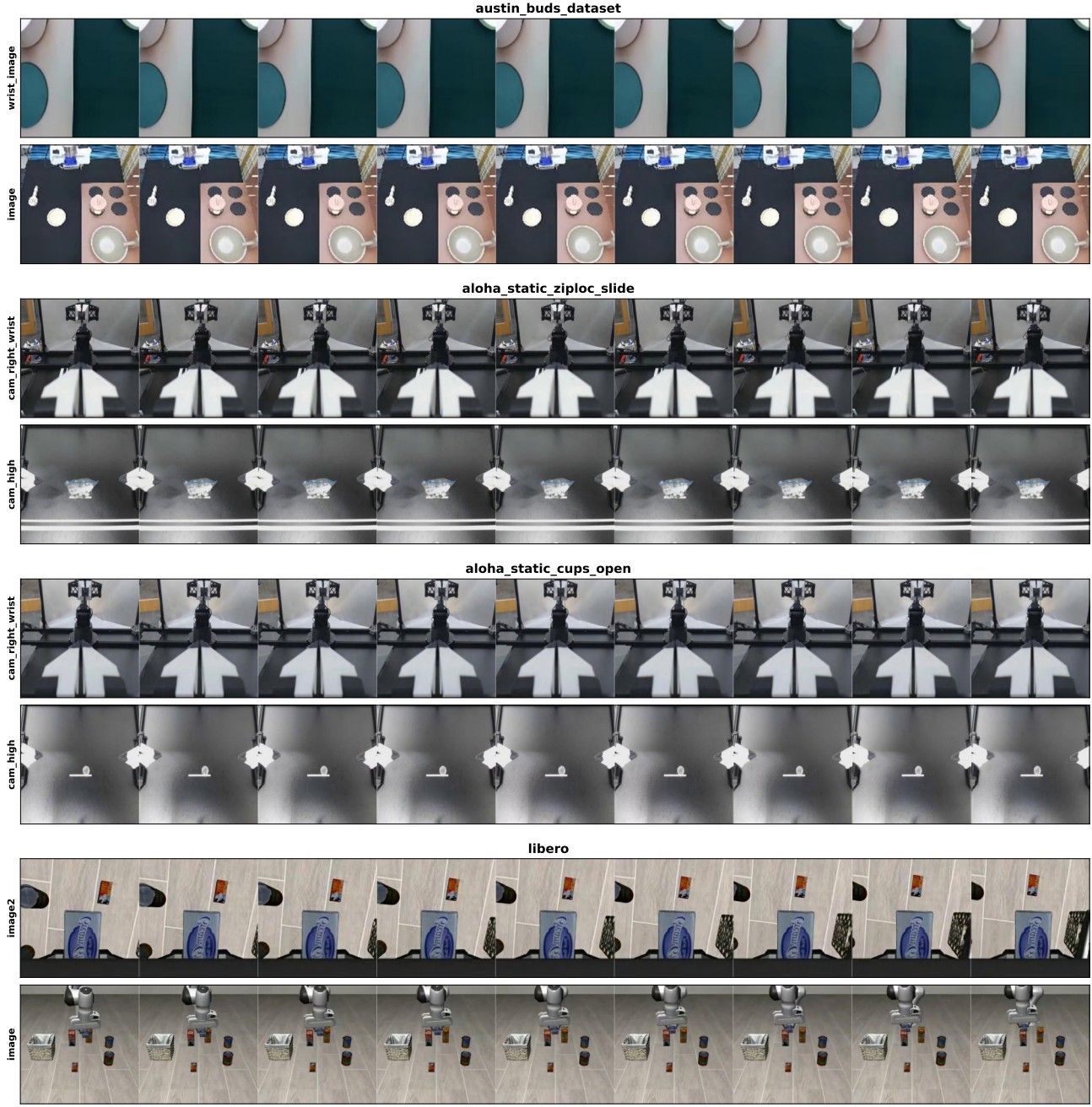

*Figure 10.* Sampled trajectories of first-person (bottom) and third-person (top) views across robotics datasets (austin_buds_dataset, aloha_static_ziploc_slide, aloha_static_cups_open, libero) from OpenX-Embodiment.

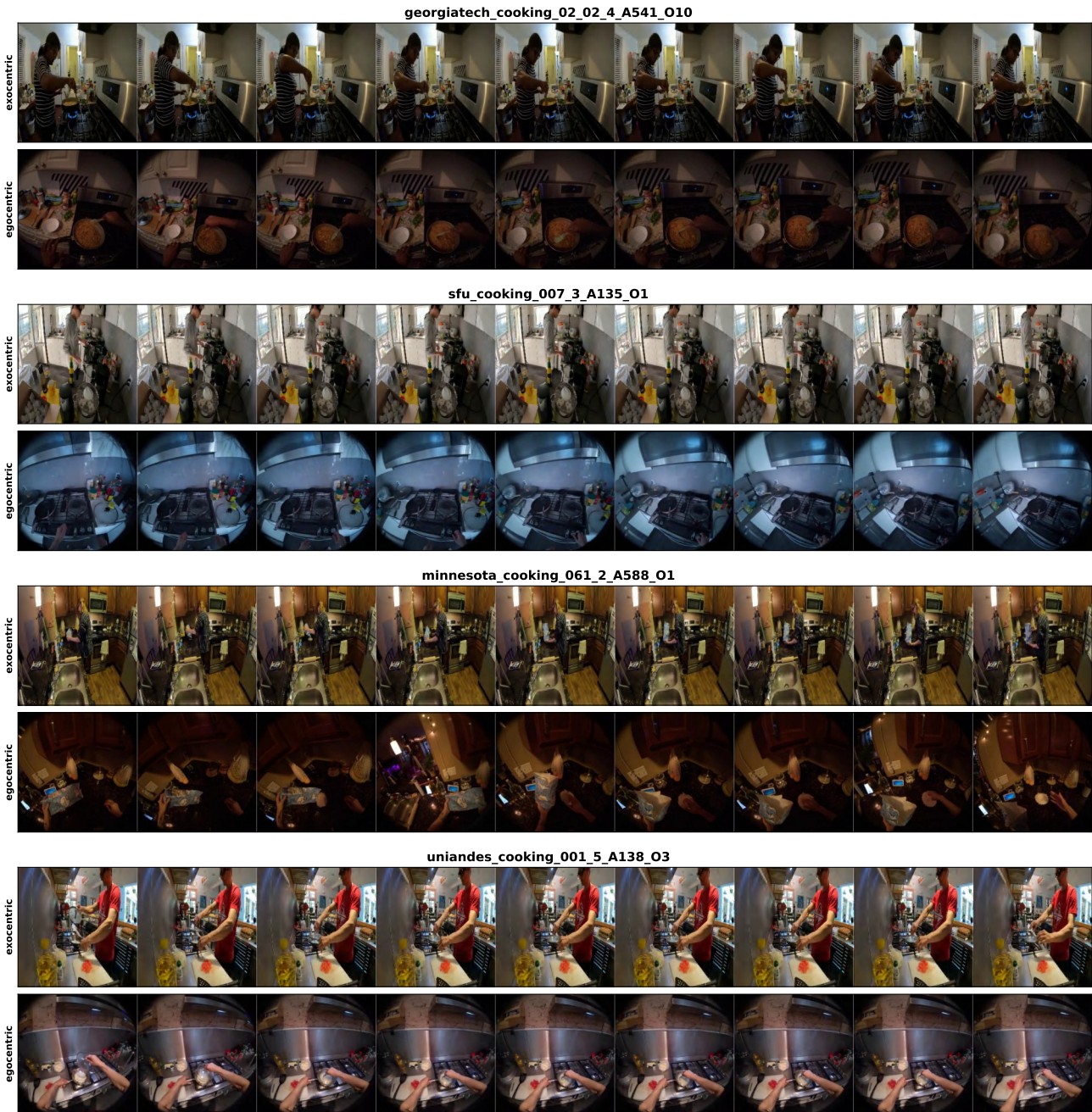

*Figure 11.* Sampled trajectories of egocentric (bottom) and exocentric (top) views across different tasks from Ego-in-Exo Perception.

*Table 6.* The Dataset Characteristics and Viewpoints

| Dataset | $|\mathcal{A}|$ | Training Viewpoints | | Additional Viewpoints | |
|---|---|---|---|---|---|
| | | Viewpoint 1 | Viewpoint 2 | Viewpoint 1 | Viewpoint 2 |
| jaco_play | 7 | image | image_wrist | – | – |
| taco_play | 7 | rgb_static | rgb_gripper | – | – |
| aloha_mobile_cabinet | 14 | cam_high | cam_left_wrist | cam_right_wrist | – |
| fanuc_manipulation | 7 | image | wrist_image | – | – |
| stanford_hydra_dataset | 7 | image | wrist_image | – | – |
| droid_100 | 7 | exterior_image_1_left | wrist_image_left | exterior_image_2_left | – |
| nyu_franka_play_dataset | 15 | image_additional_view | image | – | – |
| berkeley_autolab_ur5 | 7 | image | hand_image | image_with_depth | – |
| austin_buds_dataset | 7 | image | wrist_image | – | – |
| aloha_static_ziploc_slide | 14 | cam_high | cam_right_wrist | cam_low | cam_left_wrist |
| aloha_static_cups_open | 14 | cam_high | cam_right_wrist | cam_low | cam_left_wrist |
| libero | 7 | image | image2 | – | – |
| vp2_robosuite | 4 | image | – | – | – |
| vp2_robodesk | 5 | image | – | – | – |
| Ego-in-Exo Perception | – | egocentric | exocentric | – | – |

*Table 7.* **Detailed Comparison of Visual Planning Success Rates.** Numerical breakdown of success rates across all evaluated Robosuite and RoboDesk tasks over 100 runs.

| Method | Success Rate ↑ | | | | | | | |
|---|---|---|---|---|---|---|---|---|
| | Robosuite | Open slide | Red button | Blue button | Green button | Flat block | Upright block | Open drawer |
| Action-free | 20.25±0.89% | 2.12±1.56% | 3.77±1.23% | 26.22±10.33% | 36.27±8.46% | 0.00±0.00% | 11.96±2.88% | 3.61±2.74% |
| Raw Action | 72.25±1.92% | 2.50±2.76% | 30.83±3.63% | 84.17±11.05% | 52.50±10.64% | 0.00±0.00% | 25.00±3.73% | 11.67±5.00% |
| FICC | 60.12±15.44% | 1.12±0.13% | 13.64±7.44% | 58.92±14.07% | 56.21±15.01% | 0.00±0.00% | 3.71±0.86% | 9.10±4.28% |
| LAPO | 52.10±7.23% | 0.00±0.00% | 17.22±8.77% | 44.20±19.54% | 33.59±4.89% | 0.00±0.00% | 10.45±7.10% | 12.87±3.50% |
| AdaWorld | 64.37±3.71% | 1.30±0.56% | 22.88±2.99% | 31.22±7.12% | 8.78±3.55% | 0.00±0.00% | 6.11±1.24% | 5.97±2.81% |
| MuCoLA | 68.90±3.64% | 3.38±1.88% | 27.68±2.79% | 86.23±10.71% | 51.32±17.29% | 0.00±0.00% | 19.76±4.51% | 13.08±3.27% |
| w prior | 62.58±4.88% | 0.00±0.00% | 21.23±4.27% | 72.65±9.86% | 33.29±17.83% | 0.02±0.01% | 11.44±4.57% | 10.02±6.70% |
| w/o consistency | 38.97±9.56% | 0.00±0.00% | 12.44±5.79% | 56.20±21.53% | 33.65±7.00% | 0.00±0.00% | 9.87±4.17% | 9.72±3.59% |

# E. Additional Experimental Results

### E.1. Quantitative Results of VP$^2$

Table Section E.1 presents the detailed quantitative performance of MuCoLA, baseline models, and ablation variants. We report the mean success rates and standard deviations for each task, calculated over 100 independent evaluation runs.

### E.2. Visualization of Future Prediction

In Figures 12, 13, and 14, we visualize the future frame predictions generated by world models conditioned on latent actions inferred by MuCoLA and AdaWorld. We provide a qualitative analysis of both successful outcomes and failure cases to elucidate the comparative strengths and limitations of each method in capturing physical dynamics and visual details.

### E.3. Analysis of MuCoLA's Trajectories on VP$^2$ Benchmark

In Figure 15, we provide a visualization of the open-loop planning trajectories generated by MuCoLA across diverse tasks in the VP$^2$ benchmark. The visualizations reveal a clear dichotomy in performance based on task dynamics.

MuCoLA demonstrates robust planning capabilities in "push button" tasks characterized by discrete, impulse-based interactions, such as *Push* (Red, Blue, Green) and *Robosuite Push*. In these scenarios, the latent action space effectively captures the causal intervention required to trigger state transitions, maintaining semantic consistency throughout the rollout.

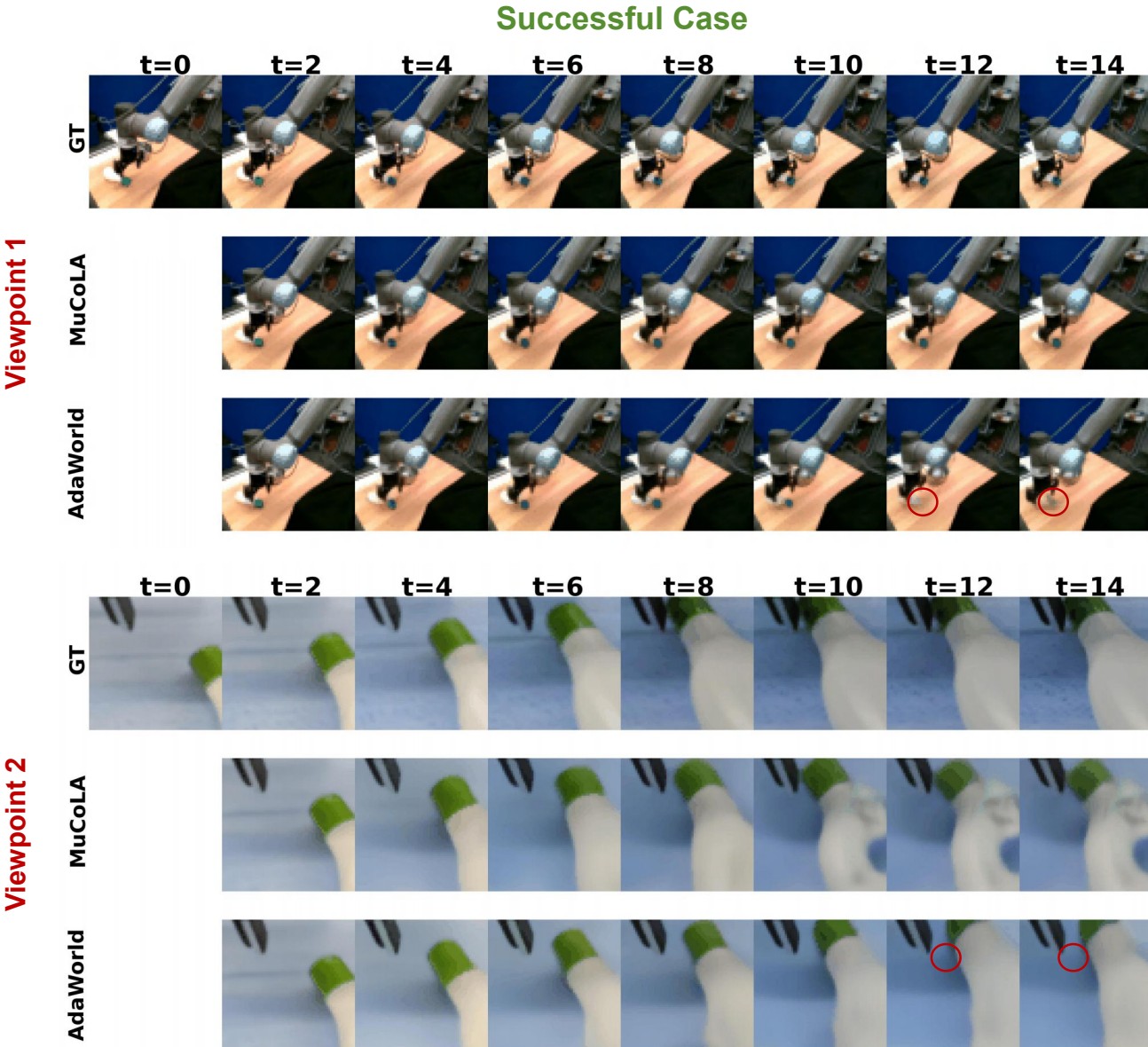

*Figure 12.* **Qualitative comparison on the *berkeley_autolab_ur5* dataset.** We compare the future frames synthesized by world models conditioned on latent actions inferred by MuCoLA and AdaWorld across two viewpoints. As highlighted by the red circles in Viewpoint 1, MuCoLA successfully anticipates the position of the target water bottle, whereas the AdaWorld baseline fails to preserve the object, resulting in the target vanishing from the prediction.

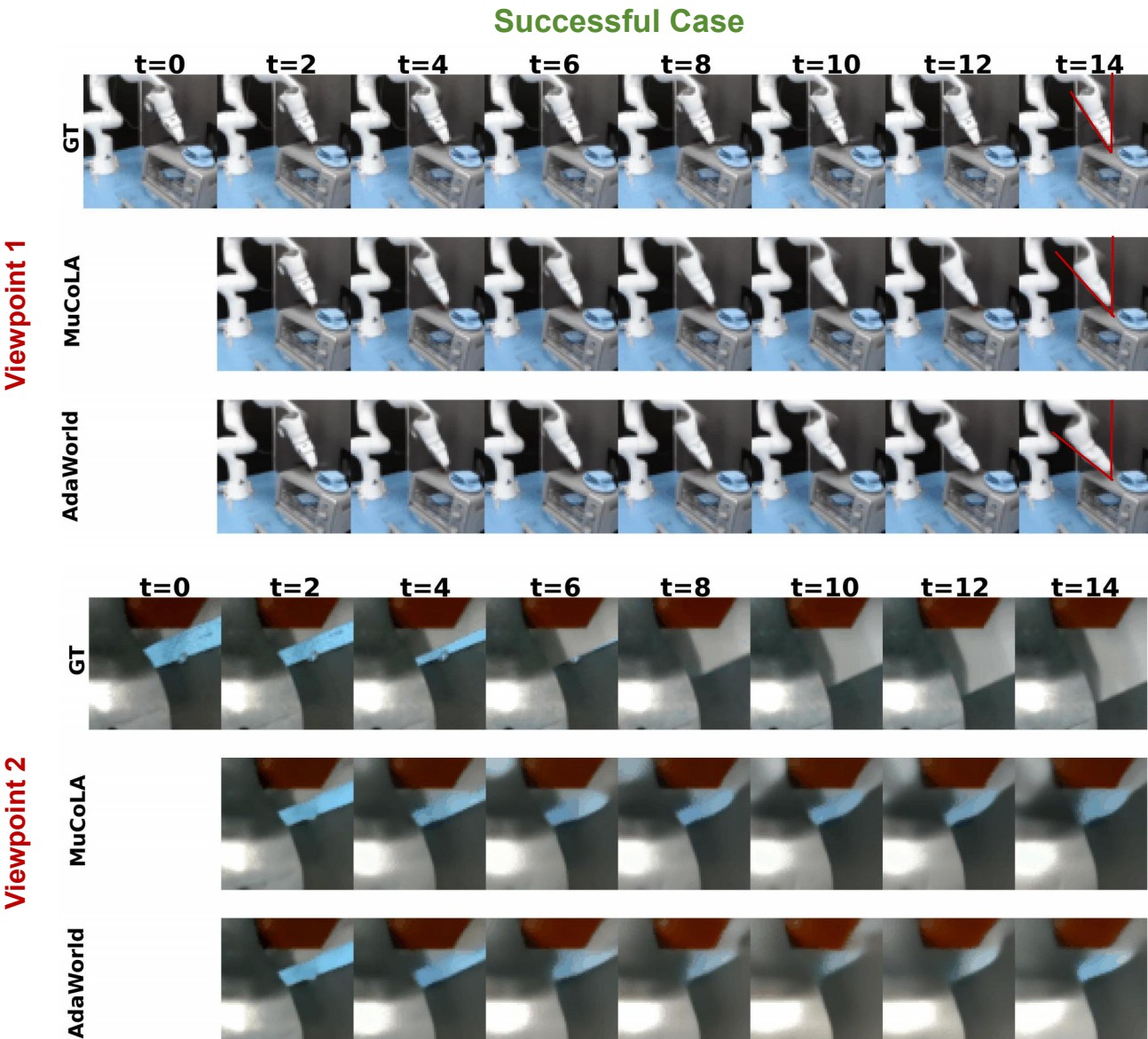

*Figure 13.* **Prediction fidelity comparison on the *stanford_hydra_dataset*.** This figure displays future frames generated by world models driven by MuCoLA and AdaWorld latent actions. The red lines in Viewpoint 1 indicate the gripper's articulation; MuCoLA predicts an opening angle that aligns closely with the ground truth, demonstrating superior fine-grained control compared to AdaWorld.

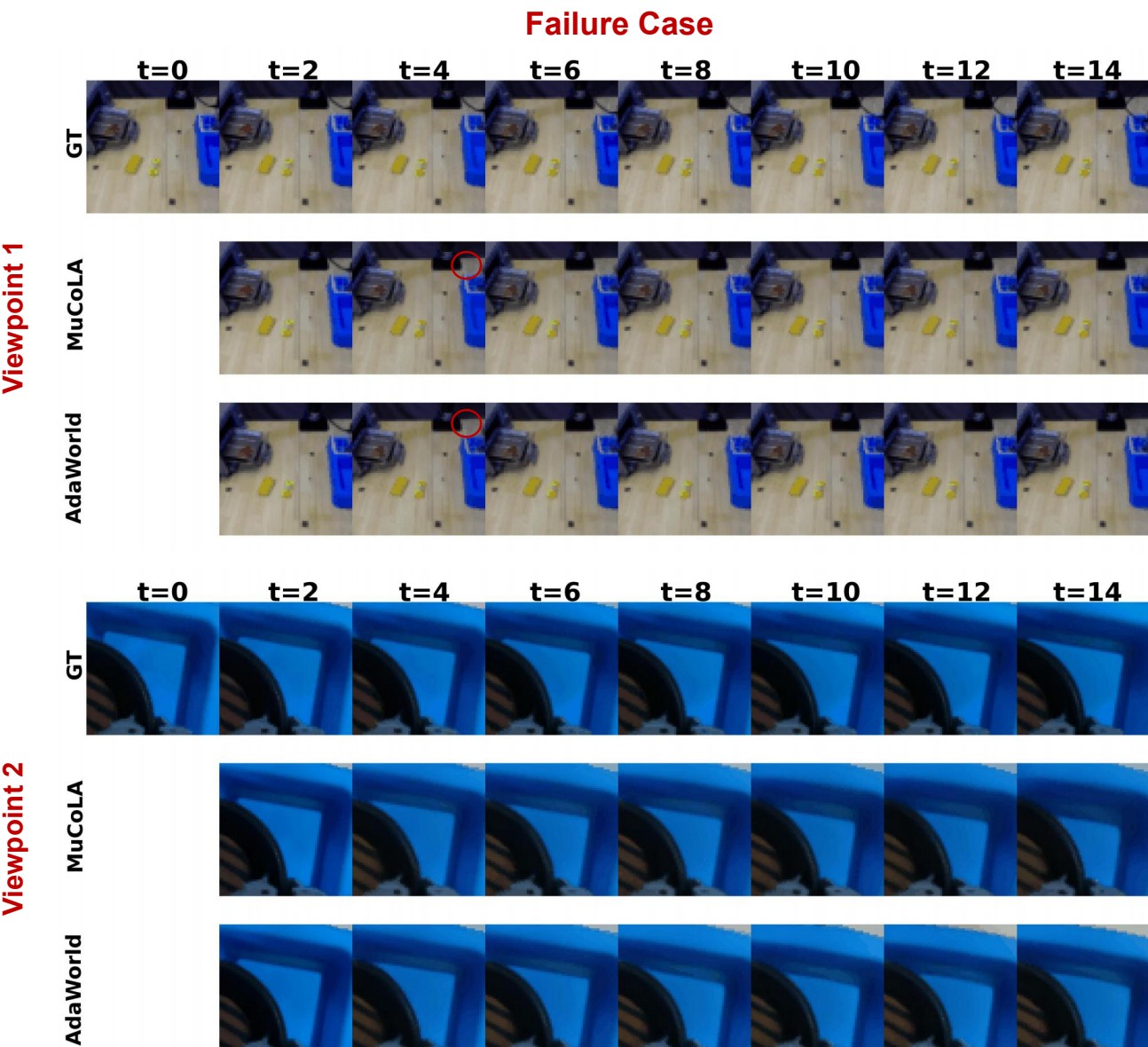

*Figure 14.* **Failure case analysis on the *jaco_play* dataset.** We examine the predictions from MuCoLA and AdaWorld to identify common limitations. As highlighted by the red circles in Viewpoint 1, both models fail to reconstruct fine-grained visual details, specifically the connecting cables on the robotic arm. This indicates a shared challenge for current methods in modeling high-frequency visual elements.

Conversely, the model exhibits limitations in long-horizon tasks necessitating precise prehensile manipulation and sustained object interaction, notably *Open Slide* and *Flat Block*. As illustrated in the failure cases, while the planner initiates the approach correctly, it struggles to maintain the geometric constraints required for continuous grasping and sliding. We attribute this to two primary factors:

- **Compounding Visual Drift:** In tasks like *Open Slide*, the visual state of the sliding mechanism changes continuously. Small errors in the autoregressive prediction accumulate over the long horizon, leading to a dissociation between the robotic end-effector and the object in the generated future.

- **Spatial Precision in Latent Space:** Successfully manipulating a flat block or a small slider requires high-frequency spatial precision. As noted in the failure case analysis (Appendix E.2), the compression into a latent action space may filter out fine-grained contact details necessary for maintaining a stable grasp over extended temporal windows.

We plan to address the aforementioned issues in future work.

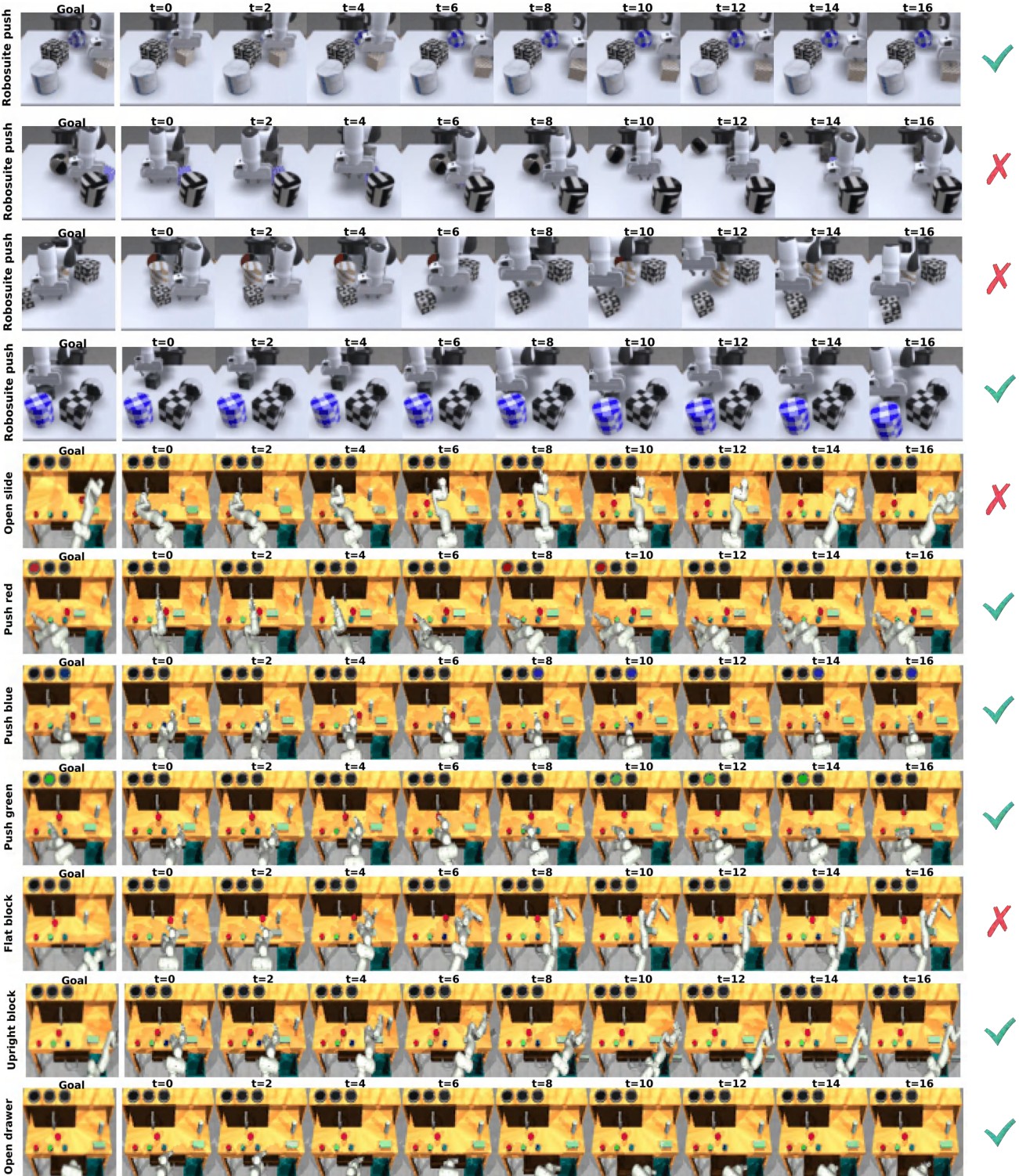

*Figure 15.* Qualitative visualization of open-loop planning trajectories generated by MuCoLA on the VP² benchmark. Rows marked with green ticks (✓) denote successful task completion, while red crosses (×) indicate failure cases. The model exhibits strong performance on short-horizon, impulse-driven tasks (e.g., *Push Button*, *Open Drawer*), but encounters difficulties in long-horizon tasks requiring sustained fine-grained manipulation (e.g., *Open Slide*, *Flat Block*), often due to spatial drift in the predicted contact dynamics.

