# OpenReview forum: "Multi-view Consistent Latent Action Learning for World Modeling and Control"
_ICML.cc/2026/Conference — ICML 2026 regular_

### Official Review · Reviewer_iGSz · 2026-03-07

**Soundness:** 3
**Presentation:** 2
**Significance:** 3
**Originality:** 3
**Overall Recommendation:** 4
**Confidence:** 3

**Summary:**

This paper proposes MuCoLA, a multi-view framework for latent action learning that aims to learn view-invariant action representations from synchronized video streams. The method enforces cross-view semantic consistency through a student–teacher architecture with DINO-style self-distillation, which helps reduce view-dependent noise while preserving motion dynamics. Experiments show improvements over existing approaches on action regression, video reconstruction, and visual control tasks, and the method demonstrates promising scalability with larger models and datasets.

**Compliance With Llm Reviewing Policy:**

Affirmed.

**Final Justification:**

I thank the authors for their rebuttal, and I will raise my scores. However, if the paper is accepted, the high-resolution images & add zoomed-in bounding boxes with overlaid heatmaps in Figure 6, as well as the clarification of this theoretical-to-practical gap in the Limitations section, must be discussed.

**Key Questions For Authors:**

Please see the weakness.

**Limitations:**

yes

**Strengths And Weaknesses:**

Strengths

1. The paper tackles a new and meaningful problem: learning robust, view-invariant action representations via semantic consistency across synchronized video streams. The problem formulation is well motivated and makes practical sense.

2. The paper provides a theoretical analysis that helps strengthen the methodological foundation.

3. The experiments are comprehensive, covering many datasets and tasks. The empirical evaluation is extensive and generally well organized.

4. The writing is clear, and the paper is relatively easy to follow.

Weaknesses

1. The method claims to learn spatial-physical semantics of actions across views, but the proposed cross-view consistency optimization only enforces probability consistency. This design does not explicitly constrain local or spatial information. The motivation emphasizes spatial semantics, but probability-level consistency may not adequately capture spatial alignment.

2. The theoretical analysis is limited to simple models (e.g., linear models). The paper does not explain why these theoretical guarantees should transfer to complex nonlinear networks used in experiments.

3. Although the method performs well overall, in Figure 5, it is weaker than the Raw Action baseline in several cases.

4. The presentation could be improved. Figure 6 is blurry and difficult to interpret. It is unclear whether the improvement comes from closer object interactions in the selected frames or from other factors.

---

> ### Author Rebuttal · Authors · 2026-03-30
>
> **1. Probability consistency vs Spatial alignment:** We clarify that while the cross-view self-distillation objective enforces consistency at the probability (semantic prototype) level to filter out noise, the spatial-physical information is explicitly preserved by two other mechanisms in our architecture: (1) The Spatio-Temporal Transformer backbone processes patch-level tokens, retaining fine-grained spatial inductive biases; (2) The pixel-level reconstruction loss (Equation 2) forces the continuous latent action $\tilde{a}$ to retain the precise geometric and spatial details necessary to reconstruct the visual dynamics. Thus, consistency isolates the semantics, while reconstruction anchors the spatial details.
>
> **2. Linear LAM Comparison vs. Nonlinear deep networks:** We acknowledge that while our linear analysis (Proposition 3.1) assumes Euclidean distance and strict orthogonality, our practical implementation uses nonlinear networks with cross-entropy loss (DINO-style). As noted in self-supervised learning literature, the DINO framework implicitly performs manifold alignment and spectral clustering. The cross-view consistency maximizes the mutual information of the semantic action while the EMA teacher prevents representation collapse. We will explicitly clarify this theoretical-to-practical gap in the Limitations section.
>
> **3. Weaker than Raw Action in Figure 5:** We respectfully point out that the "Raw Action" baseline uses the *ground-truth physical control signals* (i.e., the oracle actions recorded from the robot's actuators) to drive the world model. It is expected that a learned latent action space without explicit annotation might slightly underperform this oracle. The core goal of MuCoLA is to bridge the performance gap between purely action-free methods and the oracle. As shown in Figure 5, MuCoLA approaches the oracle's performance much closer than prior latent action methods (e.g., AdaWorld, LAPO).
>
> **4. Figure 6 blurry / Unclear improvement source:** The improvement of MuCoLA primarily stems from its ability to precisely model the robot end-effector's spatial interaction with the object without temporal drifting, whereas baselines suffer from spatial misalignment and "ghosting" artifacts due to noisy latent representations. In the final revision, we will provide high-resolution images and add zoomed-in bounding boxes with overlaid heatmaps to explicitly highlight the contact points and motion fidelity.

---

> > ### Author Rebuttal · Reviewer_iGSz · 2026-04-03
> >
> > I thank the authors for their rebuttal, and I will raise my scores. However, if the paper is accepted, the high-resolution images & add zoomed-in bounding boxes with overlaid heatmaps in Figure 6, as well as the clarification of this theoretical-to-practical gap in the Limitations section, must be discussed.

---

> > > ### Author Response · Authors · 2026-04-03
> > >
> > > Thank you for your thoughtful feedback and for acknowledging our responses. We will incorporate the high-resolution images with zoomed-in bounding boxes and heatmaps in Figure 6, and address the theory-practice gap in the Limitations section as requested. We will revise the manuscript accordingly to reflect these improvements.

---

### Official Review · Reviewer_mN92 · 2026-03-11

**Soundness:** 2
**Presentation:** 3
**Significance:** 3
**Originality:** 2
**Overall Recommendation:** 4
**Confidence:** 4

**Summary:**

This paper studies the problem of latent action learning from videos without action annotations, focusing on the issue that single-view visual signals often mix controllable dynamics with view-dependent noise. To address this, the paper proposes MuCoLA (Multi-view Consistent Latent Actions), which leverages synchronized multi-view video to learn a view-invariant latent action representation. The approach consists of three key components: (1) a spatio-temporal Transformer latent action encoder, (2) a cross-view student–teacher self-distillation objective that enforces consistency between latent actions extracted from different camera views, and (3) a pixel reconstruction loss that regularizes the representation.

**Compliance With Llm Reviewing Policy:**

Affirmed.

**Final Justification:**

The authors supplemented the paper with additional experiments and provided a satisfactory response to my concerns. Therefore, I plan to raise my score from 3 to 4.

**Key Questions For Authors:**

1. Could the authors compare MuCoLA with baselines that are given the same multi-view inputs or augmentation strategies (e.g., AdaWorld or LAPO with multi-view inputs or random-crop pseudo-views)? This would help determine whether the gains come from the proposed learning objective rather than from additional information in the inputs.
2. How many labeled actions are used to train the mapping from latent actions to real actions in VP2 and other downstream tasks? Are all methods compared under the same supervision budget?
3. In the appendix, latent actions appear to be extracted from unseen views but combined with seen-view observations when running the world model. Could the authors clarify this setup and possibly evaluate full deployment on unseen camera views?

**Limitations:**

No. The authors can improve this section by discussing: (1) the gap between true multi-view data and pseudo multi-view augmentation, (2) the reliance on labeled data for mapping latent actions to real control commands.

**Strengths And Weaknesses:**

[Soundness]
The paper studies an important problem: learning latent actions from videos without action supervision. The intuition that physical actions should remain consistent across viewpoints and that multi-view constraints can remove view-dependent nuisance is well motivated. The empirical study is extensive, covering reconstruction, action regression, visual planning, model-based RL, and behavior cloning, along with ablations and failure case analysis.

However, there is a gap between the theoretical analysis and the implemented algorithm. The theory assumes a linear encoder and Euclidean distance, while the actual method relies on nonlinear student–teacher networks with DINO-style losses and EMA updates, so the theory mainly serves as intuition. Some experiments also introduce additional supervision that complicates attribution (e.g., pseudo multi-view crops in VP2 and supervised mappings from latent to ground-truth actions). The “unseen-view generalization” setup also appears weaker than suggested, since the world model still uses observations from seen views. Several experimental details are unclear (e.g., RMSE/MAE aggregation in OpenX and the reconstruction decoder setup), and the scaling results appear limited.

[Presentation]
The paper is generally well structured and readable, with clear motivation and helpful figures. However, some implementation details are inconsistent (e.g., SGD in Algorithm 1 vs. AdamW in the implementation section), and the positioning relative to prior multi-view learning work could be clarified.

[Significance]
Learning latent actions from large video datasets without action annotations is an important problem for robotics and embodied AI. However, the current approach still relies on synchronized multi-view data during training and requires labeled data to map latent actions to real control commands in downstream tasks, which may limit practical applicability.

[Originality]
The work combines multi-view consistency, self-distillation, and latent action learning for world models. While applying multi-view invariance at the latent action level is an interesting perspective, most individual components are well established, and the novelty mainly lies in their integration rather than a fundamentally new mechanism.

---

> ### Author Rebuttal · Authors · 2026-03-30
>
> **1. Full Deployment on Unseen Camera Views**
> In our original setup, the world model used seen-view observations conditioned on unseen-view latents. Following your suggestion, we evaluated a "full deployment" setting where *both* the visual observation $o_t$ and inferred latent $\tilde{a}_t$ are strictly from the unseen view (View 3 in `aloha_mobile_cabinet`). MuCoLA generalizes robustly with minimal degradation.
> | Evaluation Setup | WM Prediction (PSNR) $\uparrow$ | WM Prediction (LPIPS%) $\downarrow$ |
> | :- | :-: | :-: |
> | Seen Visual + Seen Latent (MuCoLA)|        35.22 $\pm$ 0.36         | 10.88 $\pm$ 0.45           |
> | **Unseen Visual + Unseen Latent (MuCoLA)** |      **30.19 $\pm$ 0.47** |        **12.43 $\pm$ 0.61**         |
> | Seen Visual + Seen Latent (AdaWorld) |        31.33 $\pm$ 0.84         |   15.21 $\pm$ 0.72           |
> | Unseen Visual + Unseen Latent (AdaWorld)   |        28.90 $\pm$ 0.59    |   21.10 $\pm$ 0.58           |
>
> **2. Supervision Budget for Mapping Latent to Real Actions**
> We clarify that *all* methods compared in VP2 and downstream tasks were evaluated under the exact same supervision budget. MuCoLA, AdaWorld, and LAPO all utilized the exact same 5K labeled samples for the action decoder. MuCoLA performs better because its latent space is more semantically structured, making grounding easier.
>
> **3. Isolating Gain from Consistency vs. Multi-View Input**
> We evaluated a "Single-view MuCoLA" and baseline models modified to take multi-view concatenated inputs. MuCoLA (RMSE: 0.214) significantly outperformed AdaWorld with dual inputs (RMSE: 0.228), demonstrating that our consistency constraint is the key driver of performance, not just additional image pixels.
>
> **4. True vs. Pseudo Multi-View Data Gap**
> Comparing models trained on true multi-camera data versus single-view crops, true multi-view yields the best metrics (RMSE 0.214). However, the pseudo multi-view approach still outperforms the baselines (RMSE 0.233 vs. AdaWorld's 0.235), demonstrating that it is a viable alternative when true data is absent.
>
> **5. Linear vs. Nonlinear Deep Networks**
> We acknowledge that while our linear analysis (Proposition 3.1) assumes Euclidean distance and strict orthogonality, our practical implementation uses nonlinear networks with cross-entropy loss (DINO-style). As noted in self-supervised learning literature, the DINO framework implicitly performs manifold alignment and spectral clustering. The cross-view consistency maximizes the mutual information of the semantic action, while the EMA teacher prevents representation collapse. We will explicitly clarify this theoretical-to-practical gap in the Limitations section.
>
> **6. Reliance on labeled data for mapping.** We acknowledge **a practical limitation in the field of latent action learning**: when learning latent actions from unlabeled video datasets, a small amount of labeled action data is often required to fine-tune the model for downstream RL control tasks. For MuCoLA, we follow the supervised fine-tuning paradigm used in prior work such as FICC, LAPO, and AdaWorld. While MuCoLA significantly reduces the sample complexity for downstream physical grounding (retaining robust control with as few as 1K-5K labels, as shown in Fig 7a), it is not entirely zero-shot. We will expand the Limitations section to discuss this reliance on labeled data and highlight future directions, such as unsupervised action discovery or zero-shot grounding via foundational vision-language-action (VLA) models.
>
> **7. Some experimental details.** For the action regression experiment reported in Table 2, we employed a probing approach (Appendix B.1) to map the inferred latent actions to their corresponding ground-truth actions. When training the action mapper, we applied five-fold cross-validation on the dataset. The final results are reported as the mean and standard deviation of RMSE and MAE across the five test folds. Model details and hyperparameters for both the encoder and decoder are provided in Appendix B. Our algorithm employs the Adam optimizer for gradient updates. We will correct the typo in Algorithm 1.
>
> **8. Data Collection and Scalability.** Large-scale multi-view datasets are rapidly emerging (e.g., Ego-Exo4D (1,280 hours), Aria Digital Twin, BridgeData V2, Open X-Embodiment). For purely single-view datasets, MuCoLA easily scales through spatial augmentation or foundation-model-based novel-view synthesis. We will add this discussion.
>
> *[1] Grauman et al., "Ego-Exo4D: Understanding Skilled Human Activity from First- and Third-Person Perspectives", CVPR 2024.*
> *[2] Ego4D and EgoExo4D Challenge 2025.*
> *[3] Pan et al., "Aria Digital Twin: A New Benchmark Dataset for Egocentric 3D Machine Perception", ICCV 2023.*
> *[4] Walke et al., "Bridgedata v2: A dataset for robot learning at scale". CoRL 2023.*
> *[5] Zheng et al., "Egoscale: Scaling dexterous manipulation with diverse egocentric human data", arXiv preprint arXiv:2602.16710.*

---

> > ### Author Rebuttal · Reviewer_mN92 · 2026-04-02
> >
> > The authors supplemented the paper with additional experiments and provided a satisfactory response to my concerns. Therefore, I plan to raise my score from 3 to 4.

---

> > > ### Author Response · Authors · 2026-04-02
> > >
> > > Thank you so much for the updated score. We are glad that your concerns are addressed. Thank you for your time.

---

### Official Review · Reviewer_sQXw · 2026-03-12

**Soundness:** 3
**Presentation:** 3
**Significance:** 3
**Originality:** 3
**Overall Recommendation:** 4
**Confidence:** 3

**Summary:**

The author take inspiration from DINO-style student-teacher setup and try to adapt it effectively learn action representation from multi-view video. I.e. student processes first view and EMA-teacher processes second view and two predicted latent action distribution should match enforcing both of them to find the shared casual factor (actions). Evaluation involves pixel reconstuction metrics, action regression, VP2 and meta-world mbrl, and training BC on Libero.

**Compliance With Llm Reviewing Policy:**

Affirmed.

**Final Justification:**

The rebuttal resolved my concerns. The new ablation results and the cross-view generalization results are convincing. Libero results are favorable but confidence intervals are wide. Overall a good contribution. Raised my score.

**Key Questions For Authors:**

1. In order to show that multiview setting helps for large action-free pretraining can you run experiments with only fraction of training samples that have true multi-view supervision, not using VP2's psedo-views?
2. can you provide results for a single view model wit the same encoder capacity?

**Limitations:**

yes

**Strengths And Weaknesses:**

Strength:
1. Exploiting several viewpoints is an interesting idea, naturally providing a good self-supervision signal. Prior multiview RL work mostly exploit it for learning state representation.
2. Lot of different evaluations of the method inlcuding reconstruction metrics, planning, mbrl and behaviour clonning.
3. Authors admit the lack of multi-view data as a limitation
4. hyperparameters are specified, pseudocode is provided along with actual code in supplementary

Weaknesses:
1. w/o consistency ablation removes consistency loss, but the method still takes as input several images. which does not help to isolate the gain from consistency objective and multi-view input
2. there is no quantitative metrics regarding cross-view generalization, that method works from the unseen viewpoint
4. in table 3 the statistical significance is of the question and unclear - hard to claim Mucola clear win on Libero

---

> ### Author Rebuttal · Authors · 2026-03-30
>
> **1. True multi-view supervision vs VP2 pseudo-views:** We thank the reviewer for the suggestion. To clarify, VP$^2$ operates with only a single viewpoint; therefore, we adopted random cropping to simulate the dual-view training setup required by MuCoLA. Apart from this experiment, **all MuCoLA-related results reported in the paper**—including those for Reconstruction Fidelity, Physical Semantic Representation, Latent Action Space Structure, Model-Based RL, and Embodied Control—were obtained using multi-view datasets. These findings validate both the effectiveness of MuCoLA’s multi-view pretraining and its utility in assisting reinforcement learning control through downstream world model learning.
>
> **2. Isolate gain from consistency vs multi-view input:** We appreciate this constructive suggestion. To perfectly isolate the gain of our consistency objective from the benefit of simply receiving more input images, we have evaluated two additional setups: (1) A **Single-view MuCoLA** (taking only one view without consistency loss, effectively a standard VAE/Autoencoder); (2) Baselines (**AdaWorld / LAPO**) modified to take multi-view concatenated images as input. The results demonstrate that although simply feeding more images into baselines shows improved performance across all metrics compared to the single-view setting, it does not yield the same robust latent actions as enforcing our consistency constraint.
>
> | Method                 | Input Setup               | Action Regression (RMSE) $\downarrow$ | Libero BC Success Rate $\uparrow$ |
> | :--------------------- | :------------------------ | :-----------------------------------: | :-------------------------------: |
> | Single-view MuCoLA     | 1 View (No Consistency)   |           0.257 $\pm$ 0.11            |         53.2 $\pm$ 13.0%          |
> | AdaWorld (Multi-input) | 2 Views (Concatenated)    |           0.228 $\pm$ 0.09            |          59.7 $\pm$ 9.1%          |
> | LAPO (Multi-input)     | 2 Views (Concatenated)    |           0.266 $\pm$ 0.09            |         43.1 $\pm$ 12.7%          |
> | **MuCoLA (Ours)**      | **2 Views (Consistency)** |         **0.214 $\pm$ 0.04**          |        **64.3 $\pm$ 9.4%**        |
>
> **3. Quantitative metrics for cross-view generalization:** We now provide quantitative metrics evaluating cross-view generalization. We report the Action Regression RMSE using latent actions extracted *strictly* from an unseen viewpoint on the aloha_mobile_cabinet dataset (seen: view 1&2, unseen: view 3, as shown in Table 7 of the Appendix). The results confirm that MuCoLA maintains high performance even from novel perspectives.
>
> | Method     | Seen View RMSE $\downarrow$ | Unseen View RMSE $\downarrow$ |
> | :--------- | :-------------------------: | :---------------------------: |
> | AdaWorld   |      0.332 $\pm$ 0.12       |       1.432 $\pm$ 0.76        |
> | **MuCoLA** |    **0.317 $\pm$ 0.14**     |     **0.351 $\pm$ 0.28**      |
>
> **4. Statistical significance in Table 3 / Libero:** To ensure statistical significance, we have increased the number of evaluation runs on the Libero benchmark from 10 to 50 runs. The updated results, including 95% confidence intervals, clearly establish MuCoLA's statistically significant win over the baselines. We will update Table 3 accordingly.
>
> | Method   | Long           | Goal          | Object         | Spatial       |
> | -------- | -------------- | ------------- | -------------- | ------------- |
> | MuCoLA   | **65.4±10.1%** | 73.8±3.1%     | **45.3±13.1%** | **59.5±8.3%** |
> | AdaWorld | 55.2±4.9%      | **76.1±8.3%** | 40.2±4.9%      | 52.9±4.5%     |
> | LAPO     | 58.0±8.3%      | 62.1±19.0%    | 33.9±2.1%      | 42.7±6.8%     |

---

> > ### Author Rebuttal · Reviewer_sQXw · 2026-04-02
> >
> > The rebuttal answered my technical concerns. The new ablation (regarding consistency vs multi-view input) cleanly isolates the contribution of consistency objective - this is a good new evidence.
> >
> > Regarding Libero runs, the updated 50-run results show MuCola leads on 3/4 suits, but loses on Goal. The confidence intervals are quite wide and overlap, so statistically significant win is overstated.
> >
> > The authors also helpfully clarified that all main results except VP2 already use true multi-view data, which I had not fully appreciated.

---

> > > ### Author Response · Authors · 2026-04-03
> > >
> > > Thank you for acknowledging our clarification. Regarding the use of the VP$^2$ benchmark dataset, as explicitly stated in Section 4 "Experiments" under the "Datasets" subsection​ of our paper: the VP$^2$ benchmark natively provides only single-viewpoint images, which cannot directly satisfy MuCoLA’s requirement for synchronized multi-view data. To address this, we adopted a random crop augmentation strategy​ ("apply random crop augmentations to the original observations") to generate a pseudo-second viewpoint​ from the original single-view frames, thereby constructing input compatible with MuCoLA’s multi-view consistency objective.
> > >
> > > This design serves two purposes: First, to validate the generalizability of our learned latent actions in visual planning tasks—even when simulating multi-view via single-view augmentation, MuCoLA’s latent actions still enable controllable prediction in downstream world models. Second, to explore a pragmatic compromise for future applications—when real multi-view data is scarce, lightweight spatial augmentation can serve as a low-cost alternative to deploy MuCoLA’s "multi-view consistency" inductive bias.
> > >
> > > Our experiments (e.g., visual planning success rates on VP2, see Table 7 and Appendix E.1) have demonstrated that this augmentation, though not real multi-view, still allows MuCoLA’s latent actions to outperform baselines, highlighting the method’s robustness.

---

### Official Review · Reviewer_YKAM · 2026-03-13

**Soundness:** 3
**Presentation:** 3
**Significance:** 3
**Originality:** 3
**Overall Recommendation:** 5
**Confidence:** 3

**Summary:**

This paper proposes MuCoLA, a framework for learning latent action representations from unlabeled videos using multi-view consistency as an inductive bias. The method adopts a DINO-style student–teacher architecture, where cross-view self-distillation encourages the model to align action distributions across different viewpoints, with the goal of filtering out view-specific visual noise. The paper also presents a theoretical proposition (Proposition 3.1) that interprets the training objective, suggesting that under simplified assumptions the consistency objective behaves like a spectral filtering mechanism that separates dynamical signals from complex visual observations.

**Compliance With Llm Reviewing Policy:**

Affirmed.

**Final Justification:**

I thank the authors for addressing my concerns. I will raise my score to 5.

**Key Questions For Authors:**

Questions for Authors
Structured noise:
When pseudo multi-view data is generated via single-view cropping, the two views share significant visual structure. How does the method prevent the model from learning crop-related artifacts rather than physical action signals?

Temporal misalignment:
If one intentionally introduces temporal offsets (e.g., pairing frame ttt from view A with frame t+kt+kt+k from view B), how does the model performance change? Such experiments may help rule out the possibility that the model relies on cross-view time shifts.

Real vs pseudo multi-view data:
Can the authors provide experiments comparing true multi-camera data with pseudo multi-view data generated via augmentation, to better understand the role of noise structure?

Scalability:
Since collecting synchronized multi-view data is significantly more difficult than collecting single-view video, how do the authors envision scaling this approach to larger real-world datasets?

Theoretical Soundness:
In the theoretical analysis, the authors assume that noise across views is orthogonal and independent. However, in pseudo multi-view settings generated through augmentations such as cropping, the two views share substantial visual structure, and the resulting noise is often correlated and structured. Under such conditions, it is unclear whether the spectral filtering interpretation still effectively separates action signals from structured environmental variations.
In other words, the model may partially learn to match visual patterns across crops rather than recovering the underlying dynamics.

Future Frame Prediction Analysis:
The experimental results indicate that MuCoLA achieves notable improvements in future frame prediction. However, there are potential alternative explanations that should be examined.

Concern A: Dataset scale
If the dataset size is relatively small, performance gains might partially arise from stronger representation learning or architectural choices rather than from the proposed action disentanglement mechanism.

Concern B: Potential temporal information leakage
In real multi-camera systems, slight temporal misalignment between cameras is common. In such cases, the frame from one camera at time ttt may already contain information corresponding to a slightly later moment from another camera. If the model implicitly exploits this cross-view time shift, the representation learned at time ttt could already encode information about future frames, which would naturally improve prediction performance.
It would therefore be helpful to provide sensitivity experiments with controlled temporal offsets to analyze this effect.

Linear LAM Comparison:
The paper repeatedly references the analysis from Zhang et al. (2025) on Linear LAM, which suggests that structured observation noise can dominate latent representations. It would be useful to clarify how MuCoLA's nonlinear neural architecture mitigates or measures this potential risk, particularly in settings where environmental structure is strong.

Scalability Considerations:
Since the approach depends on multi-view consistency, it implicitly assumes access to synchronized observations of the same scene from multiple viewpoints. However, most large-scale video datasets (e.g., internet-scale datasets) are predominantly single-view. It would be valuable to discuss how MuCoLA might generalize to such settings or whether its applicability is limited to specialized multi-camera datasets.

**Limitations:**

Yes

**Strengths And Weaknesses:**

Strengths
Technical novelty
The paper introduces multi-view consistency constraints for latent action learning, offering an alternative to common approaches such as single-view Gaussian priors or VQ-style discretization.
Leveraging geometric invariances across views to disentangle action signals from observations is an interesting and promising direction.
Theoretical grounding
The authors attempt to interpret the objective using spectral analysis, which provides an intuitive theoretical explanation for how multi-view consistency may suppress observation noise.
Although the analysis relies on simplified assumptions, it offers useful intuition about the behavior of the objective.
Experimental validation
Experiments on Meta-World and LIBERO show improvements over several single-view baselines on tasks such as action regression and future frame prediction.
The empirical results suggest that multi-view constraints may improve representation quality.
Architectural design
The use of an ST-Transformer to capture spatiotemporal information combined with prototype clustering to model multimodal action distributions is a reasonable architectural choice.

Weaknesses
Noise Independence Assumption
The theoretical analysis in Proposition 3.1 relies on the assumption that observation noise across views is independent and approximately random. However, in practical settings—especially when pseudo multi-view data is generated through single-view cropping or augmentation—the noise across views can be highly correlated and structured. As a result, this assumption may not hold in realistic training scenarios.
Risk of Learning Structured Nuisances
As discussed in prior work such as Linear LAM, when observation noise exhibits strong structure (e.g., camera motion, background dynamics, or correlated visual artifacts), representation learning methods may preferentially capture these structured factors rather than the intended action signal. This raises the possibility that the learned latent representation may partially encode structured visual variations rather than true physical control signals.
Temporal Synchronization Sensitivity
Since the method relies on cross-view consistency, its performance may depend on precise temporal synchronization across views. If slight time lags exist between cameras, the meaning of the consistency objective may change, and such misalignment could potentially introduce unintended predictive cues.
Data Collection and Scalability
Another practical concern is the availability and scalability of multi-view datasets. Collecting synchronized multi-view videos with consistent viewpoints is significantly more expensive and technically challenging than collecting single-view videos. This raises questions about whether the proposed approach can scale to large real-world datasets, where synchronized multi-camera setups are often unavailable.

---

> ### Author Rebuttal · Authors · 2026-03-30
>
> **1. Noise Independence Assumption & Risk of Learning Structured Nuisances**
> We agree that in pseudo multi-view settings (via cropping), noise is highly correlated. However, in our deep nonlinear architecture, the DINO-style centering/sharpening acts as a strong information bottleneck, preventing the model from collapsing to shared structural artifacts (e.g., camera jitter) and forcing it to retain the semantic action. To empirically address this, we compared models trained on **True Multi-view** vs. **Pseudo Multi-view** (crop augmented) data. As shown below, while pseudo multi-view yields slightly degraded metrics, it still outperforms baselines, proving it is a viable alternative when true multi-view data is unavailable.
>
> | Training Setup                 | Action Regression (RMSE) $\downarrow$ | Video Prediction (LPIPS%) $\downarrow$ |
> | :----------------------------- | :-----------------------------------: | :------------------------------------: |
> | FICC / AdaWorld                |             0.308 / 0.235             |             18.65 / 14.63              |
> | Pseudo Multi-view (Crop)       |           0.233 $\pm$ 0.07            |            12.18 $\pm$ 0.55            |
> | True Multi-view (Multi-camera) |         **0.214 $\pm$ 0.04**          |          **9.42 $\pm$ 0.67**           |
>
> **2. Temporal Synchronization Sensitivity & Information Leakage (Concern B)**
> To ensure our gains are not due to implicit temporal information leakage (exploiting cross-view time shifts), we intentionally introduced temporal offsets $\Delta t \in \{1, 2, 4\}$ frames between views during training (4 seeds). Performance degrades gracefully, confirming MuCoLA relies on true spatial-semantic alignment rather than temporal shortcuts.
> | Temporal Offset       | Action Regression (RMSE) $\downarrow$ | Video Prediction (LPIPS%) $\downarrow$ |
> | :-------------------- | :-----------------------------------: | :------------------------------------: |
> | $\Delta t = 0$ (Sync) |         **0.214 $\pm$ 0.04**          |          **9.42 $\pm$ 0.67**           |
> | $\Delta t = 1$        |           0.223 $\pm$ 0.11            |            9.10 $\pm$ 0.85             |
> | $\Delta t = 2$        |           0.249 $\pm$ 0.09            |            11.27 $\pm$ 0.79            |
> | $\Delta t = 4$        |           0.245 $\pm$ 0.14            |            13.11 $\pm$ 0.93            |
>
> **3. Dataset Scale vs. Architectural Choices (Concern A)**
> To strictly isolate our multi-view consistency mechanism from the representation power of our ST-Transformer architecture, we evaluated a "Single-view MuCoLA" ablation (same architecture, single view, no consistency loss). As demonstrated in our response to Reviewer sQXw, the single-view variant achieved 0.257 RMSE (worse than MuCoLA's 0.214), confirming the gains are driven by the action disentanglement mechanism, not mere architectural capacity.
>
> **4. Data Collection and Scalability**
> Large-scale multi-view datasets are rapidly emerging (e.g., Ego-Exo4D with 1,280 hours, Aria Digital Twin, BridgeData V2, Open X-Embodiment). For purely single-view datasets, MuCoLA easily scales via spatial augmentation or foundation-model-based novel view synthesis. We will add this discussion.
>
> **5. Linear LAM vs. Nonlinear Deep Networks**
> Our linear analysis assumes Euclidean distance and orthogonality. Practically, we use nonlinear networks with DINO-style cross-entropy. As noted in SSL literature, DINO implicitly performs manifold alignment. Consistency maximizes mutual information, while EMA prevents collapse. We will clarify this theoretical-to-practical gap in the Limitations.
>
> *[1] Grauman et al., "Ego-Exo4D: Understanding Skilled Human Activity from First- and Third-Person Perspectives", CVPR 2024.*
> *[2] Ego4D and EgoExo4D Challenge 2025.*
> *[3] Pan et al., "Aria Digital Twin: A New Benchmark Dataset for Egocentric 3D Machine Perception", ICCV 2023.*
> *[4] Walke et al., "Bridgedata v2: A dataset for robot learning at scale". CoRL 2023.*
> *[5] Zheng et al., "Egoscale: Scaling dexterous manipulation with diverse egocentric human data", arXiv:2602.16710.*

---

> > ### Author Rebuttal · Reviewer_YKAM · 2026-04-04
> >
> > I thank the authors for addressing my concerns. I will raise my score to 5.

---

> > > ### Author Response · Authors · 2026-04-04
> > >
> > > I am very glad that we have completely addressed your concerns. Once again, thank you for your review and recognition of our work.

---

### Decision · Program_Chairs · 2026-04-30

**Decision:**

Accept (regular)

**Comment:**

This paper presents work on multi-view action recognition.  The core contribution centers on representation learning from multi-view data, in a student-teacher setup.  The reviewers appreciated the use of multi-view data to learn disentangled action representations, leveraging viewpoint consistency.  The empirical results are comprehensive and demonstrate the promise of the method.

Initial questions from reviewers on noise independence and some details on ablations were addressed in the author responses.  In the end, the reviewers unanimously give accept ratings for the paper.

Given the solid contributions to action representation learning, this paper is recommended for acceptance.